# Deficiency of ASGR1 promotes liver injury by increasing GP73-mediated hepatic endoplasmic reticulum stress

Zhe Zhang [1,7], Xiang Kai Leng [1,7], Yuan Yuan Zhai [1,7], Xiao Zhang[1], Zhi Wei Sun[2], Jun Ying Xiao[1], Jun Feng Lu [1], Kun Liu[3], Bo Xia[1], Qi Gao[2], Miao Jia[2], Cheng Qi Xu [4], Yi Na Jiang[5], Xiao Gang Zhang [6] ✉, Kai Shan Tao [3] ✉ & Jiang Wei Wu [1] ✉

Liver injury is a core pathological process in the majority of liver diseases, yet the genetic factors predisposing individuals to its initiation and progression remain poorly understood. Here we show that asialoglycoprotein receptor 1 (ASGR1), a lectin specifically expressed in the liver, is downregulated in patients with liver fibrosis or cirrhosis and male mice with liver injury. ASGR1 deficiency exacerbates while its overexpression mitigates acetaminophen-induced acute and CCl4-induced chronic liver injuries in male mice. Mechanistically, ASGR1 binds to an endoplasmic reticulum stress mediator GP73 and facilitates its lysosomal degradation. ASGR1 depletion increases circulating GP73 levels and promotes the interaction between GP73 and BIP to activate endoplasmic reticulum stress, leading to liver injury. Neutralization of GP73 not only attenuates ASGR1 deficiency-induced liver injuries but also improves survival in mice received a lethal dose of acetaminophen. Collectively, these findings identify ASGR1 as a potential genetic determinant of susceptibility to liver injury and propose it as a therapeutic target for the treatment of liver injury.

Liver injury, characterized by hepatocyte damage, hepatic inflammation and fibrosis, is considered the most common cause of liver disease worldwide[1,2]. It is often caused by a variety of stimuli, including pharmaceutical agents such as the clinically relevant drug acetaminophen (APAP)[3], toxins (CCl4, a common chemical reagent used to establish animal models of liver injury)[4], alcohol[5] and virus[6]. Liver injury eventually could progress into serious diseases such as cirrhosis and hepatocellular carcinoma (HCC)[7,8]. Although it is well accepted that genetic factors play crucial roles in the pathogenesis of all types of liver injuries[9–11], the precise genetic predisposing factors for liver injury remain inadequately understood.

Asialoglycoprotein receptor 1 (ASGR1) is a major subunit of ASGPR (also known as Ashwell-Morell receptor, Ashwell receptor or hepatic lectin) and is highly conserved among mammals[12,13]. It is a transmembrane protein specifically expressed in hepatocytes and plays a pivotal role in maintaining circulating glycoprotein homeostasis[14–16]. By binding to terminal galactose/N-acetylgalactosamine, ASGR1 mediates endocytosis and lysosomal degradation of circulating desialylated glycoproteins thereby participating a diverse biological functions[17–20]. A large-scale sequencing analysis of Icelanders revealed that *ASGR1* haploinsufficiency is associated with a reduced risk of coronary artery disease (CAD), initiating the exploration of ASGR1 in the development

[1]Key Laboratory of Animal Genetics, Breeding and Reproduction of Shaanxi Province, College of Animal Science and Technology, Northwest A&F University, Yangling, China. [2]Beijing Sungen Biomedical Technology Co. Ltd, Beijing, China. [3]Department of Hepatobiliary Surgery, Xi-Jing Hospital, Air Force Medical University, Xi'an, China. [4]College of Life Science and Technology, Huazhong University of Science and Technology, Wuhan, China. [5]Department of Pathology, the First Affiliated Hospital of Xi'an Jiaotong University, Xi'an, China. [6]Department of Hepatobiliary Surgery, the First Affiliated Hospital of Xi'an Jiaotong University, Xi'an, China. [7]These authors contributed equally: Zhe Zhang, Xiang Kai Leng, Yuan Yuan Zhai.
✉e-mail: little_gang17@xjtu.edu.cn; taokaishan0686@163.com; wujiangwei@nwafu.edu.cn

of cardiovascular disease (CVD)[21]. In our previous work, we generated an ASGR1-deficient pig model to investigate the potential causal relationship between ASGR1 and CVD. This model recapitulates reduced risk factors for CVD in humans, providing crucial insights into the causality between ASGR1 and the incidence of CVD[22]. During the course of this investigation, we incidentally observed liver injury under normal feeding conditions in ASGR1-deficient pigs[22], suggesting a potential correlation between hepatic *ASGR1* expression and liver injury. In line with this observation, a study on the role of ASGR1 in the lethal coagulopathy of sepsis reported increased hepatocyte death in ASGR1-deficient mice[17]. Additionally, another study showed that ASGR1 deficiency promotes LPS/galactosamine (GalN)-induced liver injury in mice[23]. Contrarily, ASGR1 deficiency has also been reported to attenuate liver injury in LPS-induced sepsis mice[24]. These results suggest that, beyond its established role in CVD, ASGR1 is implicated in liver injury and requires further investigation.

Accumulating evidence shows the pivotal role of endoplasmic reticulum (ER) stress in the initiation and progression of liver injury[3,25,26]. Sustained ER stress disrupts ER structure and/or function, leading to cell death and liver injury[25,27]. Under non-stress conditions, the ER stress marker immunoglobulin heavy chain binding protein (BIP) forms a complex with the three sensors IRE-1, PERK and ATF-6, maintaining the non-activated state of signal transduction factors[26,28]. When stressed, BIP dissociates from these sensors, triggering the unfolded protein response (UPR)[28,29]. Our previous studies demonstrated elevated hepatic ER stress in ASGR1-deficient pigs[22], implicating ER stress in ASGR1 deficiency-induced liver injury. Considering ASGR1's role in the endocytosis and degradation of circulating glycoproteins[18,30], and the documented contribution of specific glycoproteins such as Golgi Protein 73 (GP73)[31] and canopy homolog 2[32] to ER stress, it is highly plausible that ASGR1 deficiency may disrupt the balance of certain circulating glycoproteins, leading to ER stress and liver injury.

In this work, to explore the role of ASGR1 in liver injury, we generated an ASGR1-deficient mouse model, and assessed the phenotype of these mice under basal conditions, APAP-induced acute liver injury and CCl4-induced chronic liver injury. Our results revealed that ASGR1 deficiency induces spontaneous liver injury in mice under basal conditions and aggravates both APAP-induced acute and CCl4-induced chronic liver injuries. Conversely, AAV8-mediated hepatic *Asgr1* over-expression protects against both acute and chronic liver injuries. Mechanistically, we found that ASGR1 interacts with the ER stress mediator GP73 and facilitates its lysosomal degradation. Hence, ASGR1 deficiency leads to elevated circulating levels of GP73. Neutralization of GP73 not only alleviates both acute and chronic liver injuries in ASGR1-deficient mice but also improves survival in mice subjected to a lethal dose of APAP. Furthermore, we showed that in cirrhotic patients, those with decreased *ASGR1* expression exhibit increased serum GP73 levels and hepatic ER stress. Our findings indicate that ASGR1 deficiency exacerbates liver injury and the ASGR1-GP73 axis emerges as a potential therapeutic target for liver injury.

## Results

### Hepatic ASGR1 is reduced in humans and mice with liver injury
To investigate the potential association between ASGR1 and liver injury, we first determined hepatic ASGR1 expression in patients diagnosed with liver fibrosis or cirrhosis. The results showed significant reductions in both mRNA and protein levels of ASGR1 in these livers compared to normal controls (Fig. 1a–c, Supplementary Fig. S1a–e). We found an inverse correlation between hepatic *ASGR1* mRNA expression and serum levels of liver function biomarkers, including alanine aminotransferase (ALT), aspartate aminotransferase (AST), alkaline phosphatase (ALP) and gamma-glutamyl transferase (GGT), in patients with liver cirrhosis (Supplementary Fig. S2a–d). Apart from humans, in mouse models with liver injuries induced by APAP and CCl4

for acute and chronic injuries, respectively (Fig. 1d, e), hepatic mRNA and protein levels of ASGR1 were also significantly downregulated (Fig. 1f–k). Collectively, these data highlight the potential involvement of ASGR1 in liver injury.

### ASGR1-deficient mice show liver injury under basal conditions
To further investigate the role of ASGR1 in liver injury, we generated ASGR1-deficient (*Asgr1−/−*) mice using CRISPR/Cas9 technology (Supplementary Fig. S3a). The genotypes of mice were identified by PCR (WT: 659 bp; Homozygotes: 896 bp; Heterozygotes: 896 bp/659 bp) (Supplementary Fig. S3b). ASGR1 protein was undetectable in livers of *Asgr1−/−* mice (Supplementary Fig. S3c), indicating successful gene deletion. Under normal conditions, despite similar body weight and liver/body weight ratio between *Asgr1−/−* mice and their WT controls (Supplementary Fig. S4a, b), we observed significantly increased serum levels of ALT, AST, ALP and GGT in *Asgr1−/−* mice at the age of six months (Supplementary Fig. S4c–f). Morphological analysis of liver sections showed inflammatory cell infiltration in *Asgr1−/−* mice (Supplementary Fig. S4g), indicative of mild hepatic inflammation. Consistent with this, the mRNA expression of proinflammatory cytokines *Tnf-α, Il-6, Mcp1,* and *Il-1β* was increased in livers of *Asgr1−/−* mice (Supplementary Fig. S4h). In addition, the mRNA expression of pro-apoptosis genes *Bax, Caspase3* and *Caspase9* was increased, while the mRNA expression of anti-apoptosis marker *Bcl-2* was decreased in livers of *Asgr1−/−* mice (Supplementary Fig. S4i). Together, these results show that ASGR1 deficiency causes mild liver injury under basal conditions.

### ASGR1 deficiency aggravates acute and chronic liver injuries in mice
Following the observation of mild liver injury in *Asgr1−/−* mice under basal conditions, we wondered whether external stresses would exacerbate these injuries. To explore this, *Asgr1−/−* mice and WT controls were treated with APAP to induce acute liver injury. *Asgr1−/−* mice displayed higher serum levels of ALT and AST than WT controls (Supplementary Fig. S5a, b), accompanied with a marked increase in areas of hepatocyte necrosis (Supplementary Fig. S5c). These results suggest that ASGR1 deficiency accelerates APAP-induced acute liver injury.

To further investigate the role of ASGR1 in chronic liver injury, we treated mice with CCl4 for 6 weeks to induce chronic liver injury (Fig. 2a). *Asgr1−/−* mice exhibited significantly increased serum levels of ALT, AST, ALP and GGT compared to WT controls, despite similar body weight and liver/body weight ratio (Fig. 2b–d, f–h). Following CCl4 treatment, *Asgr1−/−* mice displayed a greater prevalence of hepatic ballooning degeneration and inflammatory cell infiltration compared to WT controls (Fig. 2e, i), accompanied with increased mRNA expression of proinflammatory cytokines *Tnf-α, Il6, Mcp1* and *Il-1β* (Fig. 2j). TUNEL staining showed higher cell apoptosis in livers of ASGR1-deficient mice compared with WT controls (Fig. 2e, k). The mRNA expression of pro-apoptosis markers *Bax, Caspase3* and *Caspase9* was increased, while the mRNA expression of anti-apoptosis marker *Bcl-2* was decreased in livers of *Asgr1−/−* mice (Fig. 2l). In addition, Sirius red staining and Masson's trichrome staining showed notably increased fibrotic areas in *Asgr1−/−* mice (Fig. 2e, m, o). Hepatic mRNA expression of profibrotic genes *α-Sma, Col1a1, Timp1* and *Tgf-β* was increased in *Asgr1−/−* mice (Fig. 2n). Additionally, the content of hydroxyproline was significantly increased in livers of *Asgr1−/−* mice, indicating prominently increased collagen deposition in the absence of ASGR1 (Fig. 2p). Since both APAP and CCl4 induced liver injuries require metabolic activation by cytochrome P450 2E1 (CYP2E1)[33], we examined the influence of ASGR1 on CYP2E1 expression and found unaffected levels in *Asgr1−/−* mice with either treatment (Supplementary Fig. S6a–f). Together, these results show that ASGR1 deficiency accelerates both acute and chronic liver injuries in mice.

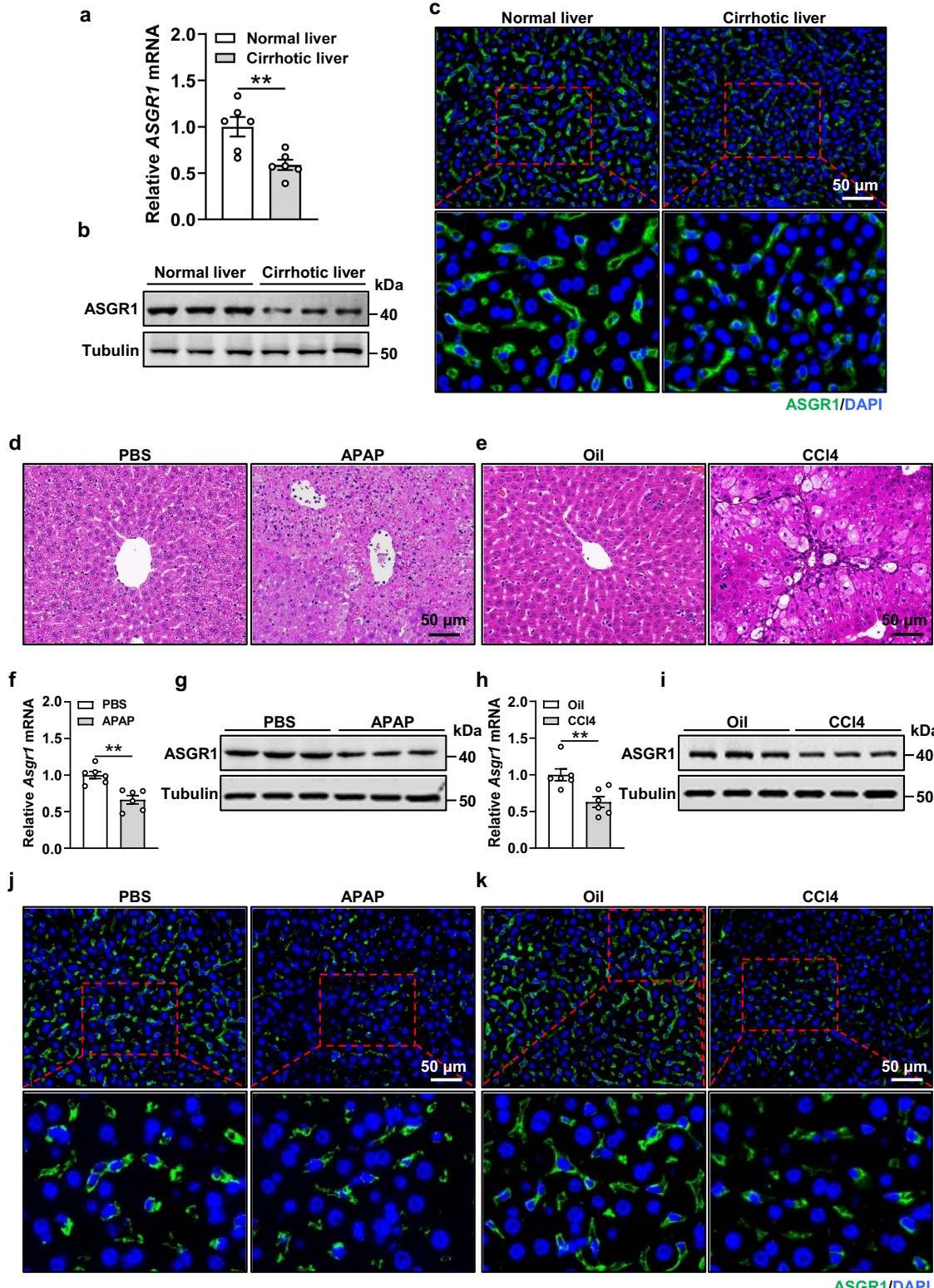

**Fig. 1 | Hepatic ASGR1 is downregulated in cirrhotic patients and liver injured mice. a**, **b** Relative mRNA and protein expression of asialoglycoprotein receptor 1 (ASGR1) in normal and cirrhotic human liver tissues ($n = 6$ per group). **c** Immunofluorescence staining of ASGR1 in normal and cirrhotic human liver tissues. **d**–**k** 8-week-old mice were intraperitoneally injected with either acetaminophen (APAP, 400 mg/kg body weight, a single dose) to induce acute liver injury with PBS as control treatment, or carbon tetrachloride (CCl4, 1 ml/kg body weight, twice a week for 6 weeks) to induce chronic liver injury with oil as control treatment ($n = 6$ per group). **d**, **e** H&E staining of liver sections. Scale bar, 50 μm. **f**–**i** Relative mRNA and protein expression of hepatic ASGR1 in mice treated with APAP or CCl4. **j**, **k** Representative immunofluorescence staining of ASGR1 in livers of mice treated with APAP or CCl4. Scale bar, 50 μm. Data are presented as mean ± SEM. *P* values were calculated by two-tailed unpaired *t*-test. *$P < 0.05$, **$P < 0.01$. Source data are provided as a Source data file.

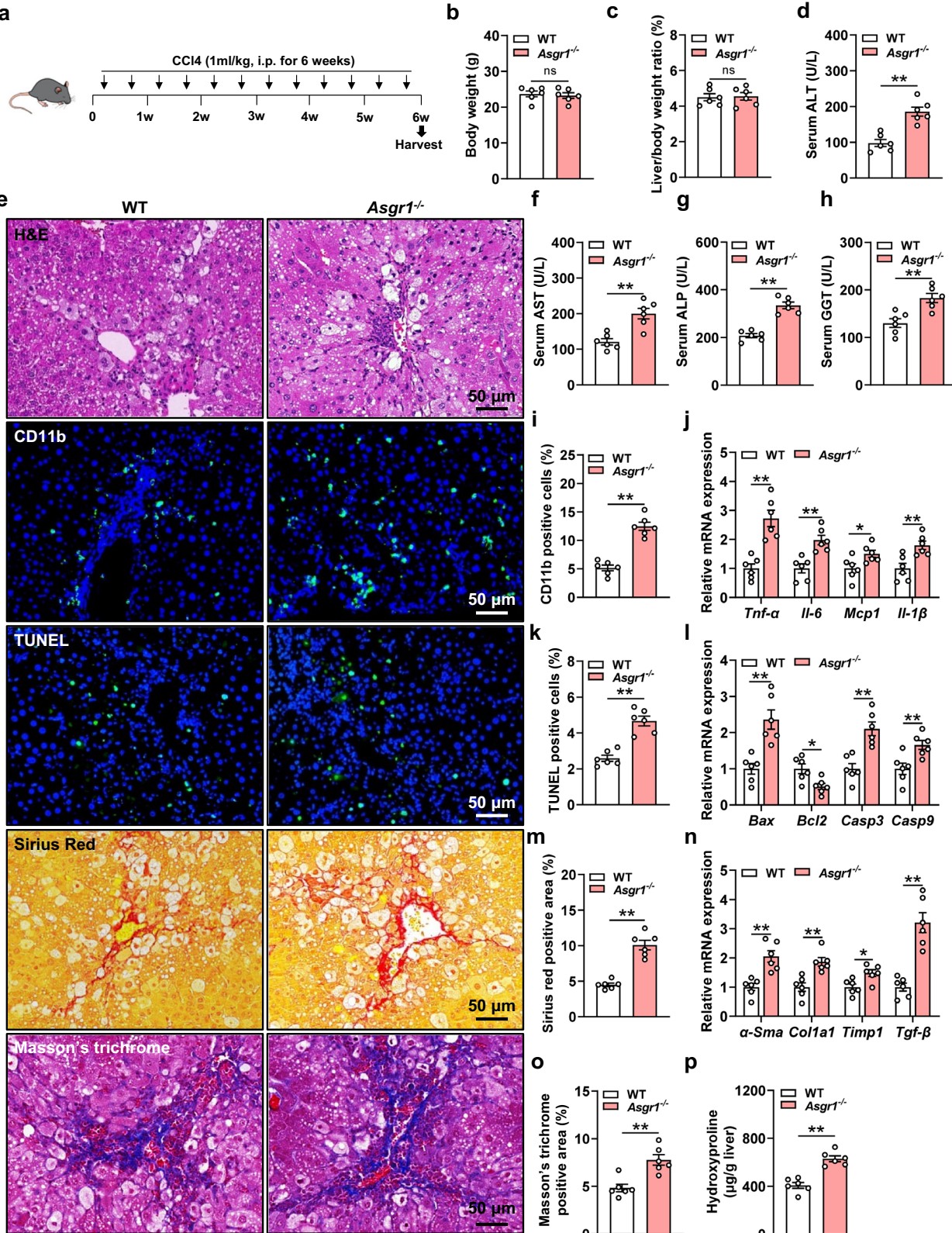

## Hepatic overexpression of *Asgr1* protects against acute and chronic liver injuries in mice

We further investigated the effect of ASGR1 on liver injury by hepatic overexpression (OE) of *Asgr1* in mice. Mice were injected with AAV8-*Asgr1* (*Asgr1*-OE mice) or control AAV8-NC via tail vein and subsequently treated with CCl4 for 6 weeks (Fig. 3a). AAV8-*Asgr1* markedly increased hepatic protein levels of ASGR1 (Supplementary Fig. S7a),

with no changes in the body weight and liver/body weight ratio (Fig. 3b, c). Compared with control mice, *Asgr1*-OE mice showed low levels of ALT, AST, ALP and GGT upon CCl4 treatment (Fig. 3d, f–h). Histological assessment revealed less inflammatory cell infiltration in livers of *Asgr1*-OE mice compared to AAV8-NC controls (Fig. 3e, i). In line with this, hepatic mRNA expression of genes associated with proinflammation (*Tnf-α*, *Il6*, *Mcp1* and *Il-1β*) was lower in *Asgr1*-OE mice

**Fig. 2 | ASGR1 deficiency exacerbates CCl4-induced chronic liver injury.**
**a** Schematic diagram of mice treatment. 8-week-old *Asgr1⁻/⁻* and WT control mice were intraperitoneally injected with CCl4 (1 ml/kg body weight, twice a week for 6 weeks) (*n* = 6 per group). **b, c** Body weight and liver-to-body weight ratio. **d** Serum levels of ALT. **e** H&E staining of liver sections and immunofluorescence staining of CD11b, as well as TUNEL staining, Sirius red staining and Masson's trichrome staining of liver sections. Scale bars, 50 μm. **f** Serum levels of AST (*P* = 0.001). **g** Serum levels of ALP (*P* = 0.000013). **h** Serum levels of GGT (*P* = 0.0027). **i** Quantification of immunofluorescence staining of CD11b (*P* = 0.0000083).

**j** Relative hepatic mRNA expression of the proinflammatory genes *Tnf-α, Il6, Mcp1* and *Il-1β*. **k** Quantification of TUNEL staining (*P* = 0.000069). **l** Relative hepatic mRNA expression of the apoptosis-related genes *Bax, Bcl2, Casp3* and *Casp9*. **m** Quantification of Sirius red staining (*P* = 0.000014). **n** Relative hepatic mRNA expression of the profibrotic genes *α-Sma, Col1a1, Timp1* and *Tgf-β*. **o** Quantification of Masson's trichrome staining. **p** Hepatic hydroxyproline content (*P* = 0.000066). Data are presented as mean ± SEM. *P* values were calculated by two-tailed unpaired *t*-test. \**P* < 0.05, \*\**P* < 0.01. Source data are provided as a Source data file.

than in controls with CCl4 treatment (Fig. 3j). The number of apoptotic cells and mRNA expression of pro-apoptosis genes were also significantly decreased in livers of *Asgr1*-OE mice (Fig. 3e, k, l). *Asgr1* overexpression prominently decreased collagen deposition in mice as shown by Sirius Red staining, Masson's trichrome staining and hydroxyproline quantification (Fig. 3e, m, o, p). Consistently, hepatic mRNA expression of profibrotic genes *α-Sma, Col1a1, Timp1* and *Tgf-β* was decreased in *Asgr1*-OE mice (Fig. 3n). Similarly, *Asgr1* overexpression markedly ameliorated APAP-induced acute liver injury (Supplementary Fig. S7b–d). These data indicate that hepatic overexpression of *ASGR1* ameliorates both acute and chronic liver injuries in mice.

## ASGR1-deficient mice show elevated hepatic ER stress
To explore the underlying mechanism of ASGR1-mediated liver injury, we performed RNA-seq analysis in livers of WT and *Asgr1⁻/⁻* mice and identified 2309 differentially expressed genes (DEGs) (Fig. 4a). Gene ontology analysis showed the enrichment of 69 DEGs associated with the response to ER stress in *Asgr1⁻/⁻* mice (Fig. 4b). RT-qPCR and Western blot validation confirmed a significant increase in the expression of five common ER stress markers (BIP, ATF4, ATF6, IRE1 and CHOP) in livers of *Asgr1⁻/⁻* mice (Fig. 4c, d). In livers of cirrhotic patients, decreased hepatic *ASGR1* mRNA expression was associated with increased transcript levels of ER stress markers (Supplementary Fig. S8). Collectively, these results demonstrate that ASGR1 deficiency activates hepatic ER stress.

We next tried to address the relationship between ASGR1-mediated liver injury and ER stress. *Asgr1⁻/⁻* and WT mice subjected to CCl4-induced chronic liver injury were treated with vehicle or tauroursodeoxycholic acid (TUDCA) (Fig. 4e), a pharmacological inhibitor of ER stress[34]. TUDCA treatment significantly decreased hepatic mRNA and protein levels of ER stress markers BIP and CHOP especially in *Asgr1⁻/⁻* mice (Supplementary Fig. S9a–c). Concurrently, TUDCA treatment dramatically reduced serum levels of ALT and AST in *Asgr1⁻/⁻* mice (Fig. 4f, g), mitigating ASGR1 deficiency-induced hepatic inflammation and fibrosis (Fig. 4h). The significantly increased hepatic mRNA and protein levels of genes associated with fibrosis, as well as increased hydroxyproline content in vehicle-treated *Asgr1⁻/⁻* mice, were absent in TUDCA-treated *Asgr1⁻/⁻* mice (Supplementary Fig. S9d–i). Likewise, the beneficial effect of TUDCA on liver injury was also shown in *Asgr1⁻/⁻* mice treated with APAP, as evidenced by decreased serum levels of ALT and AST (Supplementary Fig. S10a–c), along with reduced areas of hepatocyte necrosis (Supplementary Fig. S10d). These results suggest that ER stress mediates the effect of ASGR1 deficiency on liver injury in mice.

## ASGR1 physically interacts with an ER-stress mediator GP73
Considering the established role of ASGR1 in binding to terminal galactose/N-acetylgalactosamine of desialylated glycoproteins and subsequently facilitating their endocytosis/lysosomal degradation[35], we first performed immunoprecipitation (IP) of HA-tagged ASGR1 protein complexes in HepG2 cells (Supplementary Fig. S11). Subsequently, the immunoprecipitates were analyzed via LC-MS/MS (Supplementary Fig. S11). Candidate proteins were further screened using

an online tool (https://www.uniprot.org/) to identify characteristic structural features associated with ASGR1 ligand, specifically those containing terminal non-reducing galactose residues or N-acetylgalactosamine residues of desialated tri or tetra-antennary N-linked glycans, as shown by previous studies[36,37]. The integrated results identified three proteins PLVAP, CREB3L1 and GP73 as potential ligands for ASGR1 (Supplementary Fig. S11). We then performed endogenous Co-IP assay in HepG2 cells and found protein-protein interactions between ASGR1 and GP73, as well as PLVAP (Fig. 5a). Between the two candidates, GP73 was shown to interact with the ER stress marker BIP, thereby activating ER stress signaling and inducing liver injury in mice[31]. To validate this interaction, a Co-IP assay was conducted in HepG2 cells overexpressing both ASGR1 and GP73, showing a physical interaction between the two proteins (Fig. 5b, c). These results suggest that GP73 emerges as the most likely potential target of ASGR1 in liver injury. Confocal analysis further demonstrated the colocalization of ASGR1 and GP73 in HepG2 cells (Fig. 5d). Furthermore, molecular mapping assay revealed that the 161-291 amino-acid-sequence (aa) domain of ASGR1 is responsible for the direct interaction with GP73 (Fig. 5e) and the 56-205aa domain of GP73 was identified as the region responsible for binding to ASGR1 (Fig. 5f). Previous studies have identified the significance of Gln240, Trp244, and Glu253 within the carbohydrate recognition domain (CRD) of ASGR1 for its carbohydrate binding[30]. To further elucidate the interaction, we examined the interaction of HA-ASGR1 or its triple Ala-mutant (3A) with Flag-GP73 in HepG2 cells. The results showed that HA-ASGR1, but not its triple Ala-mutant, exhibited binding capability with GP73 (Fig. 5g). Given that GP73 contains two N-glycosylation sites, Asn109 and Asn144, located within the 56-205aa domain[38], we mutated these asparagines (N) to alanines (A) to explore the ASGR1-GP73 binding site. GP73-Asn144A failed to bind ASGR1, highlighting the critical role of Asn144 in the interaction between GP73 and ASGR1 (Fig. 5h). These results provide compelling evidence supporting the physical binding between ASGR1 and GP73.

## ASGR1 facilitates endocytosis and lysosomal degradation of GP73 in a clathrin-dependent manner
Given that GP73 is a candidate target of ASGR1 and is highly expressed in patients with liver cirrhosis or HCC[39], we first examined their relationship in these patients and observed negative correlations between elevated circulating GP73 levels and reduced *ASGR1* mRNA expression (Supplementary Figs. S12 and S13a). To further investigate their relationship, we measured circulating GP73 in *Asgr1⁻/⁻* and hepatic *Asgr1*-OE mice treated with CCl4 or APAP, respectively. Serum levels of GP73 were markedly increased in *Asgr1⁻/⁻* mice (Supplementary Fig. S13b, c) while decreased in *Asgr1*-OE mice (Supplementary Fig. S13d, e) compared to their corresponding controls. Considering the established role of ASGR1 in the mediation of endocytosis and lysosomal degradation of serum glycoproteins[35], we reasoned that ASGR1 may promote the degradation of glycoprotein GP73 in the circulation. To examine the endocytosis and degradation of GP73, we added red fluorescent-labeled recombinant GP73 to HepG2 cells and observed increased intracellular fluorescence intensity over time, reaching the

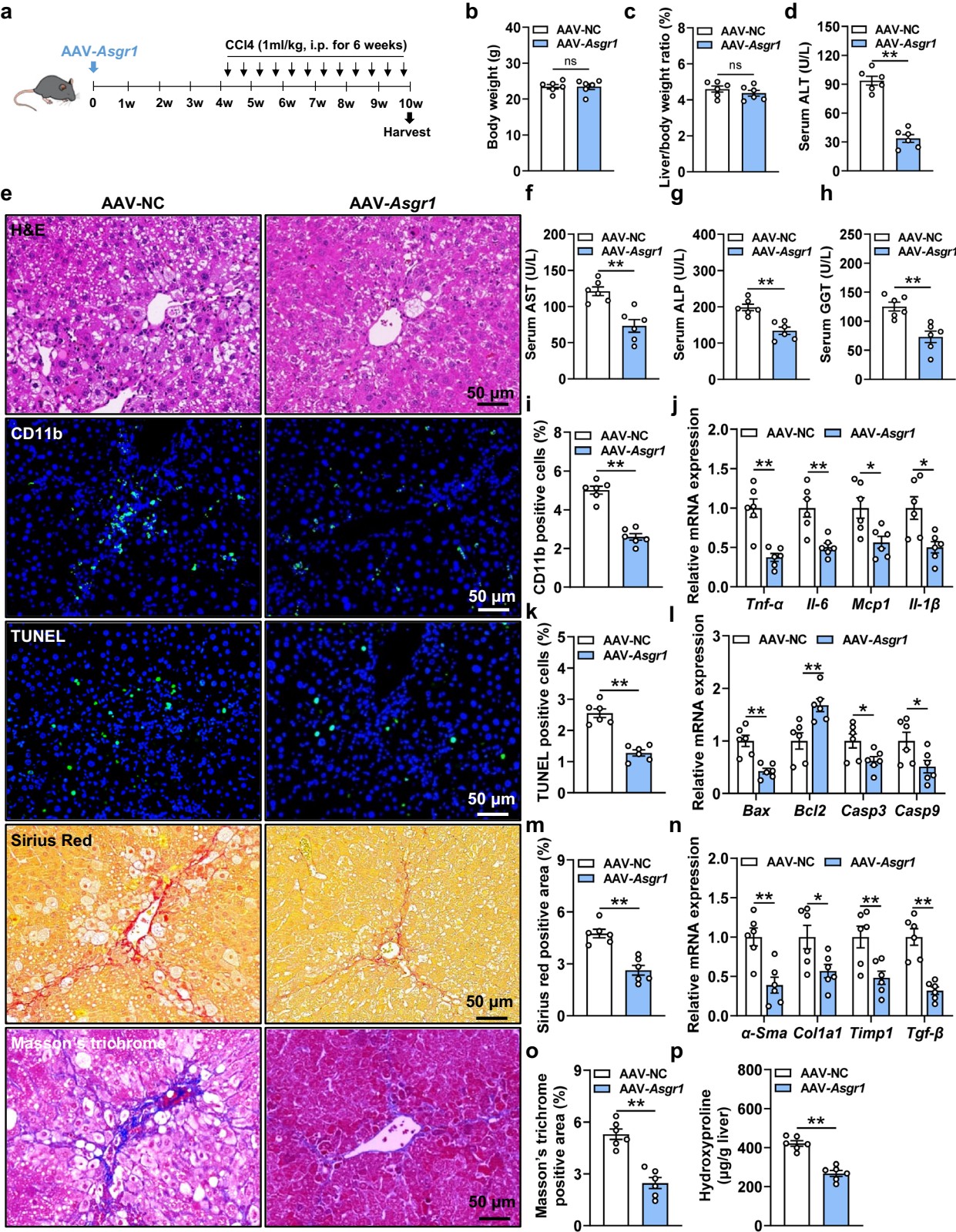

peak intensity at 6 hours post treatment (Fig. 6a). Compared with controls, *ASGR1* overexpression significantly enhanced the fluorescence intensity in HepG2 cells incubated with recombinant GP73 (Fig. 6b, c). In contrast, the presence of Asialofetuin A, a high-affinity natural ligand for ASGR1, significantly decreased the fluorescence intensity of GP73 in *ASGR1*-overexpressing cells (Fig. 6b, c). These results demonstrate that ASGR1 promotes the endocytosis of GP73.

Since clathrin plays an important role in the endocytosis of various cargo molecules[40], we proceeded to investigate whether clathrin is involved in ASGR1-mediated endocytosis of GP73 in hepatocytes. Knockdown of *clathrin heavy chain* (*CHC*) significantly reduced the fluorescence intensity of GP73 in *ASGR1*-overexpressing cells (Fig. 6d, e), indicating the essential role of clathrin in ASGR1-mediated endocytosis of GP73.

**Fig. 3 | Hepatic overexpression of Asgr1 ameliorates CCl4-induced chronic liver injury. a** Schematic diagram of mice treatment. 8-week-old mice were treated with adeno-associated virus-Asgr1 (AAV-Asgr1) or adeno-associated virus-negative control (AAV-NC) (*n* = 6 per group). After 4 weeks, mice were intraperitoneally injected with CCl4 (1 ml/kg body weight, twice a week for 6 weeks). **b, c** Body weight and liver-to-body weight ratio. **d** Serum levels of ALT (*P* = 0.0000017). **e** H&E staining of liver sections and immunofluorescence staining of CD11b, as well as TUNEL staining, Sirius red staining and Masson's trichrome staining of liver sections. Scale bars, 50 μm. **f–h** Serum levels of AST, ALP and GGT. **i** Quantification of immunofluorescence staining of CD11b (*P* = 0.0000037). **j** Relative hepatic mRNA expression of the proinflammatory genes Tnf-α, Il6, Mcp1 and Il-1β. **k** Quantification of TUNEL staining (*P* = 0.000023). **l** Relative hepatic mRNA expression of the apoptosis-related genes Bax, Bcl2, Casp3 and Casp9. **m** Quantification of Sirius red staining. **n** Relative hepatic mRNA expression of the profibrotic genes α-Sma, Col1a1, Timp1 and Tgf-β. **o** Quantification of Masson's trichrome staining (*P* = 0.000076). **p** Hepatic hydroxyproline content (*P* = 0.00002129). Data are presented as mean ± SEM. *P* values were calculated by two-tailed unpaired *t*-test. *P < 0.05, **P < 0.01. Source data are provided as a Source data file.

We next investigated whether ASGR1 mediates the degradation of GP73. In *ASGR1*-overexpressing hepatocytes, colocalization of GP73 with the lysosome marker LAMP1 was significantly enhanced (Fig. 6f, g). Treatment with Chloroquine, a lysosome-specific inhibitor, significantly blocked the lysosomal localization of GP73 in *ASGR1*-overexpressing hepatocytes (Fig. 6f, g), indicating that ASGR1 facilitates the lysosomal degradation of GP73. Knockdown of *CHC* almost completely abrogated the lysosomal localization of GP73 in *ASGR1*-overexpressing hepatocytes (Fig. 6h, i), reinforcing the notion that ASGR1 promotes the lysosomal degradation of serum GP73 in a clathrin-dependent manner. To explore whether ASGR1 is also involved in the release of GP73 into circulation, we knocked down *CHC* in HepG2 cells to prevent the interference of ASGR1-mediated GP73 endocytosis on medium GP73 content. We observed that *ASGR1* overexpression had no discernible effect on the release of GP73 (Supplementary Fig. S14a, b). Previous studies have shown that GP73 undergoes cleavage by furin before being released into circulation[31]. Consistent with this, increased GP73 secretion was observed upon furin overexpression in GP73-overexpressing HepG2 cells (Supplementary Fig. S14c, d). However, the expression of furin in *ASGR1*-overexpressing cells was not significantly different from that in controls (Supplementary Fig. S14e, f). Altogether, these data indicate that ASGR1 is involved in the endocytosis and lysosome degradation of GP73.

Based on previous findings that GP73 interacts with BIP at the plasma membrane to activate intracellular ER stress signaling and induce liver injury[31], we sought to investigate whether ASGR1 regulates their interaction and, subsequently, contributes to liver injury. Exogenous Co-IP assays showed that *ASGR1* knockdown increased the interaction between GP73 and BIP in their overexpressing hepatocytes (Supplementary Fig. S15a), while *ASGR1* overexpression reduced the binding of GP73 to BIP in these cells (Supplementary Fig. S15b). Immunofluorescence staining for GP73, BIP, and the plasma membrane marker Na+/K+-ATPase in HepG2 cells revealed a significant increase in the colocalization of GP73 and BIP at the plasma membrane upon *ASGR1* knockdown, whereas this colocalization was markedly reduced upon *ASGR1* overexpression (Supplementary Fig. S15c). Collectively, these data show that ASGR1 deficiency inhibits the endocytosis and lysosomal degradation of GP73, thereby elevating the interaction between BIP and GP73 at the plasma membrane, leading to the activation of hepatic ER stress.

### Neutralization of GP73 attenuates liver injury in *Asgr1⁻/⁻* mice

To further explore the role of GP73 in ASGR1 deficiency-mediated liver injury, we employed a neutralizing antibody to neutralize circulating GP73 in mice with chronic CCl4 treatment (Fig. 7a). Compared with IgG treatment, GP73 neutralization effectively counteracted the elevated serum levels of ALT and AST in *Asgr1⁻/⁻* mice treated with CCl4 (Fig. 7b, c). Additionally, GP73 neutralization significantly attenuated increased hepatic collagen deposition (Fig. 7d), and reduced elevated mRNA (Supplementary Fig. S16a–d) and protein (Supplementary Fig. S16e) levels of profibrotic genes in *Asgr1⁻/⁻* mice treated with CCl4. The significantly increased hydroxyproline content in *Asgr1⁻/⁻* mice was attenuated by GP73 neutralization (Supplementary Fig. S16f). We then tested the effect of GP73 neutralization on ASGR1 deficiency-mediated liver injury in mice after 10 h of APAP treatment (Fig. 7e). We chose this time point (10 h after APAP intoxication) to administer GP73 neutralizing antibody because APAP overdose patients often present late in the hospital[41], and the only FDA-approved standard antidote for APAP intoxication, N-acetylcysteine (NAC), is effective in the early stages (within 8 h)[42,43]. GP73 neutralization markedly ameliorated liver injury in *Asgr1⁻/⁻* mice with APAP intoxication, as evidenced by reduced serum levels of ALT and AST (Fig. 7f, g), as well as decreased areas of hepatocyte necrosis (Fig. 7h). Together, these results showed that neutralization of GP73 effectively attenuates liver injury in *Asgr1⁻/⁻* mice.

To unravel the mechanism by which anti-GP73 mitigates liver injury in ASGR1-deficient mice, we assessed hepatic ER stress levels and observed significant reductions in hepatic mRNA and protein levels of BIP and CHOP upon GP73 neutralization in *Asgr1⁻/⁻* mice (Supplementary Figs. S16g–i, S17). Previous studies have well established that ER stress impairs liver regeneration in partial hepatectomy, toxin- or drug induced liver injuries[44]. We then explored whether GP73 neutralization-induced reduction in ER stress contributes to enhanced liver regeneration and thus alleviates liver injury. *Asgr1⁻/⁻* mice treated with APAP displayed markedly impaired liver regeneration, characterized by downregulated expression of *cyclin A2/B1/D1/E1* and a decreased number of Ki67 positive cells when compared to their corresponding WT controls (Supplementary Fig. S18). In contrast, anti-GP73 significantly increased liver regeneration in both genotypes of mice, with a more pronounced effect observed in *Asgr1⁻/⁻* mice treated with APAP (Supplementary Fig. S18). However, when these mice were further treated with the ER stress agonist tunicamycin (Tm), the GP73 neutralization-induced liver regeneration completely disappeared in *Asgr1⁻/⁻* mice (Supplementary Fig. S19), indicating that GP73 neutralization promotes liver regeneration in *Asgr1⁻/⁻* mice by inhibiting hepatic ER stress. Together, these data provide strong support that neutralization of GP73 mitigates liver injury in *Asgr1⁻/⁻* mice.

### Neutralization of GP73 improves survival in *Asgr1⁻/⁻* mice received severe overdose of APAP

Apart from inducing liver injury, a severe overdose of APAP can lead to liver failure and death[45]. To investigate the involvement of ASGR1 and GP73 in this process, we treated mice with a lethal dose of APAP (650 mg/kg body weight) and found markedly reduced levels of hepatic *Asgr1* mRNA and elevated circulating GP73 (Supplementary Fig. S20a, b). We then treated mice with this dose of APAP and survival monitoring revealed a faster death in *Asgr1⁻/⁻* mice (died within 36 h) than in WT controls (died within 48 h) (Fig. 8a, b). To further assess whether GP73 plays a role in this process, *Asgr1⁻/⁻* mice and WT controls were given either IgG or anti-GP73 treatment (Fig. 8a). Remarkably, neutralization of GP73 significantly improved survival in both genotypes of mice compared to their respective IgG-treated groups (Fig. 8b). The survival rate of ASGR1-deficient mice increased to 70% (Fig. 8b). Thus, ASGR1 deficiency aggravates lethal APAP intoxication and targeting GP73 emerges as a promising therapeutic strategy for mitigating APAP poisoning.

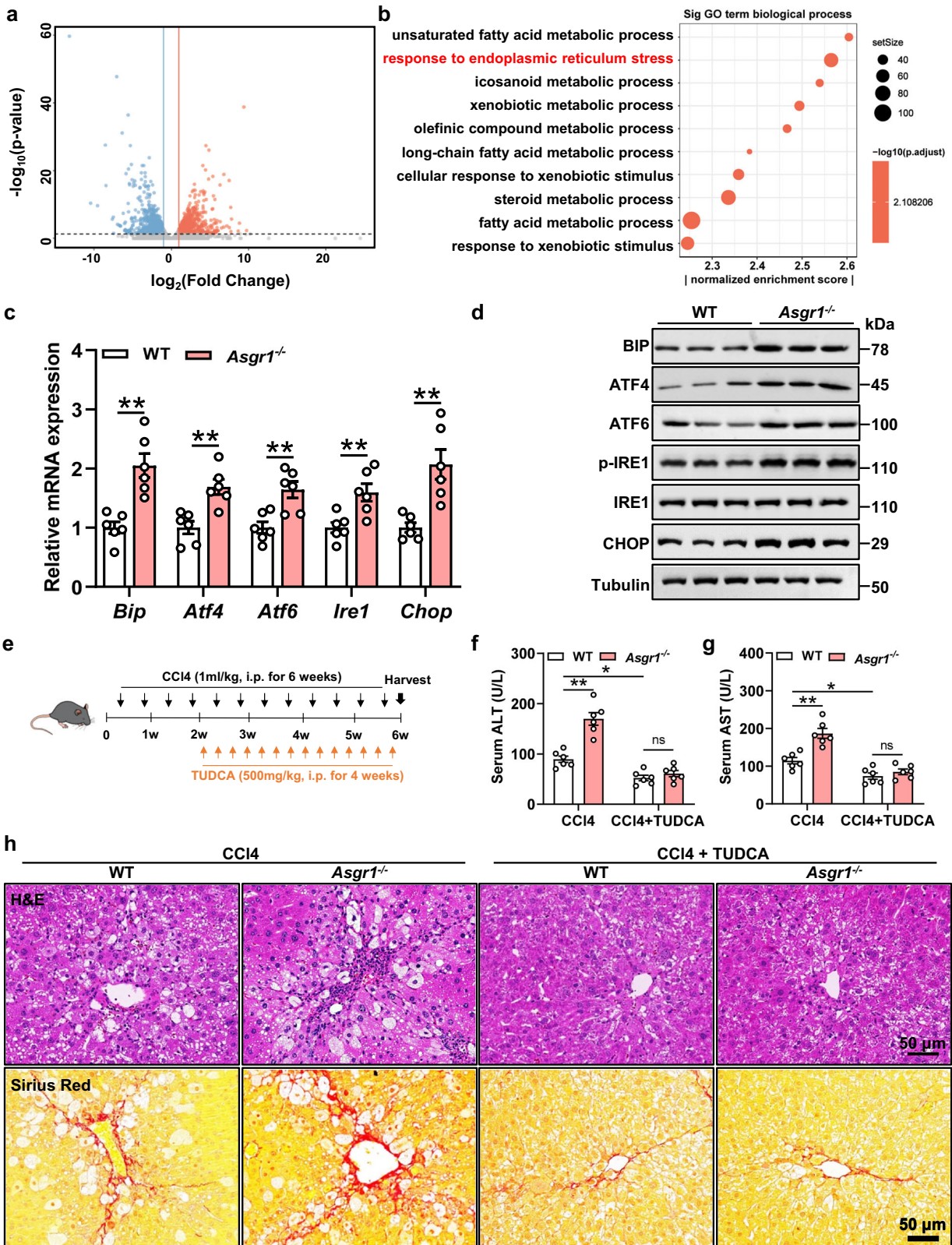

## Discussion

The beneficial impact of *ASGR1* haploinsufficiency in protecting against CAD was first documented through a large-scale GWAS analysis in Icelanders[21] and then confirmed in the UK-Biobank and the CARDIoGRAMplusC4D with three novel identified *ASGR1* variants[46]. Despite the observed cardioprotective effects, our previous investigation revealed liver injury in ASGR1-deficient pigs[22]. In this study, we found reduced hepatic *ASGR1* expression in patients with liver fibrosis or cirrhosis and in mice exhibiting liver injury. We show that ASGR1 deficiency induces spontaneous liver injury and exacerbates both acute and chronic liver injuries in mice. Mechanistically, ASGR1 physically interacts with the ER stress mediator GP73, a circulating glycoprotein and established serum marker for liver diseases such as HBV, HCC and liver cirrhosis[31,47,48], thereby facilitating its degradation.

**Fig. 4 | ASGR1-deficient mice show elevated hepatic ER stress.** RNA sequencing was performed on livers of *Asgr1⁻/⁻* mice and WT controls at 6 months. **a** Volcano plot representation of significantly up- and downregulated genes. **b** Gene Ontology (GO) analysis of significantly changed genes in biological processes. The top 10 enriched GO terms were shown in **b**. **c, d** Relative hepatic mRNA and protein expression of genes related to ER stress signaling pathways in *Asgr1⁻/⁻* and WT mice (BIP, ATF4, ATF6, IRE1 and CHOP) (*n* = 6 per group). **e** Schematic diagram of mice treatment. 8-week-old *Asgr1⁻/⁻* mice and WT controls were intraperitoneally injected with CCl4 (1 ml/kg body weight, twice a week for 6 weeks). During the last 4 weeks, mice were received either an ER stress inhibitor TUDCA (500 mg/kg body weight, every two days) or vehicle (*n* = 6 per group). **f** Serum levels of ALT (CCl4+WT *vs.* CCl4+*Asgr1⁻/⁻*, *P* = 0.0000053; CCl4+WT *vs.* CCl4+TUDCA + WT, *P* = 0.020; CCl4+TUDCA + WT *vs.* CCl4+TUDCA+*Asgr1⁻/⁻*, *P* = 0.87). **g** Serum levels of AST (CCl4+WT *vs.* CCl4+*Asgr1⁻/⁻*, *P* = 0.0002; CCl4+WT *vs.* CCl4+TUDCA + WT, *P* = 0.0309; CCl4+TUDCA + WT *vs.* CCl4+TUDCA+*Asgr1⁻/⁻*, *P* = 0.8308). **h** H&E staining and Sirius red staining of liver sections. Scale bars, 50 μm. Data are presented as mean ± SEM. Data were analyzed using Wald test (**a**) and hypergeometric test (**b**). *P* values were calculated by two-tailed unpaired *t*-test (**c**), or two-way ANOVA with Tukey's multiple comparison test (**f, g**). \**P* < 0.05, \*\**P* < 0.01. Source data are provided as a Source Data file.

ASGR1 deficiency induces liver injury by activating the GP73-mediated ER stress pathway (Fig. 8c). Neutralization of GP73 alleviates liver injury in *Asgr1⁻/⁻* mice and improves survival of mice treated with a lethal dose of APAP. We also found that decreased hepatic *ASGR1* expression correlates with increased serum levels of GP73, hepatic ER stress, and biomarkers of liver injury in patients with liver cirrhosis. These findings reveal the dual functionality of ASGR1: while its deficiency reduces the risk of cardiovascular disease, it concurrently promotes liver injury. These results corroborate that *ASGR1* is a candidate underlying genetic predisposition to liver injury.

APAP is the most common cause of drug-induced liver injury, and studies have demonstrated several genes such as Galpha (12)[49], Stard1[3] and Akr7a1[50] play critical roles in its related liver injury. CCl4 is commonly utilized to establish models of toxin-induced liver injury[4]. Here, we investigated the role of ASGR1 in liver injury induced by these two noxious stimuli and concluded that ASGR1 deficiency exacerbates the two forms of liver injury. Apart from these factors, virus infection and alcohol abuse represent two major contributors to liver injury, leading to viral hepatitis and alcoholic hepatitis, respectively[5,6]. In viral liver injury, ASGR1 has been implicated in mediating the entry of hepatitis C virus structural proteins into human hepatocytes[51] and the binding of the hepatitis B surface antigen to HepG2 cells[52]. This suggests a potential role of ASGR1 in virus-induced liver injury, and highlighting an avenue for future research. Furthermore, a previous study demonstrated that ASGPR deficiency exacerbated alcohol-induced hepatocyte apoptosis in mice[53], indicating a potential role of ASGPR in alcohol-induced liver injury. Extensive assessment of ASGR1's role in alcohol-induced liver injury warrants further investigation. It has been reported that ASGPR deficiency in mice increased hepatic inflammatory cell infiltration and hepatocyte apoptosis in response to LPS/GalN-induced liver injury[23]. Similarly, increased hepatocyte death was observed in *Asgr1⁻/⁻* mice during an investigation of the endogenous ligands of ASGR1 in sepsis[17]. Consistent with these results, we found similar phenotypes i.e. increased hepatic inflammatory cell infiltration and hepatocyte apoptosis. Together, these studies consistently support the crucial roles of ASGR1 in liver injuries induced by various stimuli.

It is worth noting that a recent study by Wang et al. [20]. elegantly demonstrated that ASGR1 deficiency in mice reduces lipid levels by promoting cholesterol excretion without inducing liver injury. The discrepancy between their findings and ours might be attributed to several factors: (1) Dietary differences. Wang's study utilized a high-fat (HF)/high-cholesterol (HC)/bile-salt (BS) diet, whereas our study employed a standard chow diet. Dietary composition significantly influences the gut microbiota, a well-accepted determinant impacting intestinal, hepatic, and systemic health[54]. Consequently, liver injury in mice could vary across microbiota[55,56]. (2) Different gene targeting strategies. Although both studies applied CRISPR/Cas9 gene targeting technology, the cleavage sites within the *Asgr1* gene differed between the two studies, resulting in genetically non-identical mouse models. We propose that future studies should incorporate multiple loss-of-function approaches across various genetic models to enhance our understanding. However, since increased cholesterol excretion was shown in *Asgr1⁻/⁻* mice by Wang et al. [20], this per se raises concerns about the potential risk of gallstones and cholestatic liver injury[57,58]. A comprehensive assessment of liver injury in *Asgr1⁻/⁻* mice over the long term, particularly under a HF/HC/BS diet, is essential to conclude the impact of ASGR1 deficiency on liver injury. Furthermore, a recent study in AAV-9 mediated *Asgr1* knockdown mice found that ASGR1 promotes liver injury by regulating monocyte-to-macrophage differentiation during LPS-induced sepsis[24]. Given that ASGR1 is predominantly expressed in hepatocytes and widely used as a hepatocyte-specific target for drug delivery[15,59], its involvement in the regulation of monocyte-to-macrophage differentiation may require cell-cell interaction. Also, systemic inflammation and multiple organ damage/failure, typical characteristics of sepsis[60], were induced in these *Asgr1* knockdown mice, adding complexity to the interpretation. Thus, several affecting factors may contribute to these conflicting results and further studies using the same mouse model or under comparable stress conditions such as sepsis are necessary.

ASGR1 is known for its role in removing and degrading potentially harmful circulating glycoproteins, although only a limited number of ligands have been identified[18,19]. Documented endogenous ligands of ASGR1 include von Willebrand factor, platelets[17], and low-density lipoprotein receptor (LDLR)[30]. Here, we present the evidence that ASGR1 physically interacts with the circulating glycoprotein GP73 and facilitates its lysosomal degradation. ASGR1 deficiency induces liver injury by activating the GP73-mediated ER stress pathway. We further demonstrate that neutralizing GP73 alleviates the exacerbated liver injury in ASGR1-deficient mice. It is worth noting that GP73 neutralization also improves the survival of ASGR1-deficient mice in the context of APAP poisoning.

Although NAC is the FDA-approved standard antidote for APAP intoxication, its efficacy is confined to a limited therapeutic window (within 8 h)[41]. Effective treatments for patients with late-stage APAP intoxication are still lacking. In this study, we showed that GP73 neutralization holds promise as a potentially effective treatment for various liver injuries and APAP poisoning induced death. Our findings reveal the translational potential of GP73 neutralization in mitigating liver injury and improving survival following APAP poisoning. This discovery paves the way for many clinical trials targeting GP73 for the treatment of APAP poisoning and other types of liver injury.

GP73 has been shown to induce ER stress activation through interaction with BIP at the plasma membrane[31] and regulate HCC growth and metastasis[61]. Here, we uncover the underlying mechanism by which the ASGR1-GP73-BIP axis activates ER stress. This regulatory mechanism may have broader implications for a range of diseases characterized by elevated circulating GP73 levels, including non-alcoholic fatty liver[62], HBV[48] and HCV[63] infections, as well as HCC[39]. Moreover, a recent study reported elevated plasma GP73 levels in patients with SARS-CoV-2 and concluded that GP73 acts as a glucogenic hormone contributing to SARS-CoV-2-induced hyperglycemia[64]. Although ASGR1's role in this process has not been investigated, it is known that ASGR1 serves as an alternative functional receptor for SARS-CoV-2 entry in various cell types[65]. Therefore, exploring whether ASGR1 regulates GP73 in the context of SARS-CoV-2-induced

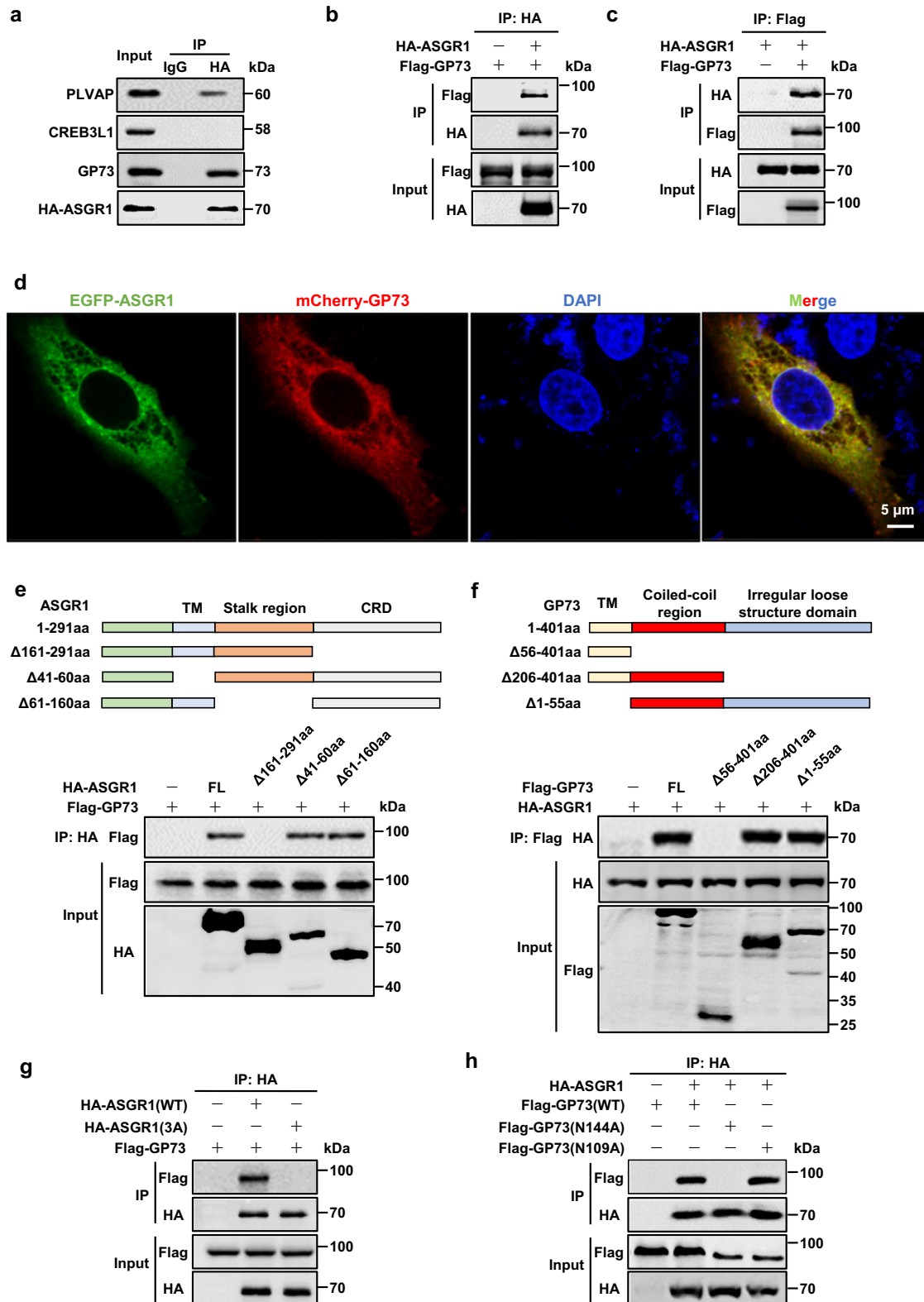

**Fig. 5 | ASGR1 physically interacts with the ER stress mediator GP73.**
**a** Endogenous Immunoprecipitation (IP) assay examining the interaction between three candidate proteins and ASGR1 in HepG2 cells, *n* = 3 biologically independent samples. **b**, **c** Exogenous Co-IP assays assessing the interaction of ASGR1 and GP73 in HepG2 cells expressing HA-tagged ASGR1 and Flag-tagged GP73, *n* = 3 biologically independent samples. **d** Immunofluorescence analysis showing colocalization of ASGR1 and GP73 in HepG2 cells expressing EGFP-ASGR1 and mCherry-GP73. Scale bars, 5 μm, *n* = 3 biologically independent samples. **e**, **f** Schematics of the ASGR1 and GP73 full-length and fragment constructs (upper panel), and Co-IP

assays analyzing the interaction domains of ASGR1 and GP73 (lower panel), *n* = 3 biologically independent samples. **g** Co-IP assays assessing the interaction of ASGR1 (WT) or mutant ASGR1 (3A) with GP73 in HepG2 cells. ASGR1 (3 A), the three residues Gln240, Trp244 and Glu253 of ASGR1 were mutated to alanine, *n* = 3 biologically independent samples. **h** Co-IP assays assessing the interaction of GP73 (WT), GP73 (N109A) and GP73 (N144A) with ASGR1 in HepG2 cells. Two asparagines (N) of GP73 in 109 and 144 were mutated to alanine, *n* = 3 biologically independent samples. Source data are provided as a Source data file.

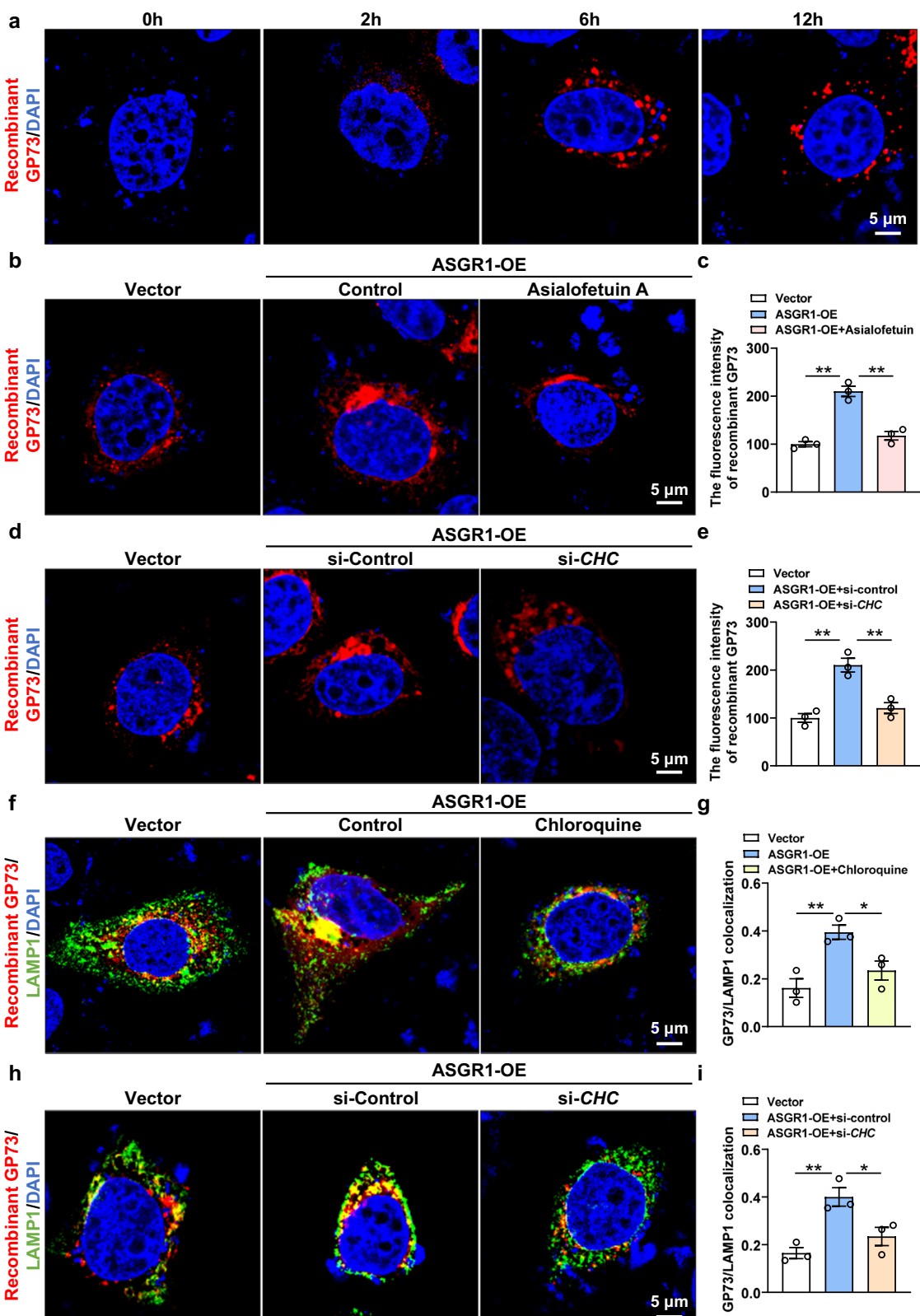

hyperglycemia merits further investigation. Together, the ASGR1-GP73 axis holds promise as a potential therapeutic target for liver injuries induced by various factors and other related diseases such as COVID-19.

While we identified ASGR1 as a potential target for the development of therapeutic strategies in liver injury, our current work has certain limitations. First, given the intricate nature of liver injury

pathogenesis, involving uncontrolled chronic inflammation and fibrosis[1,2], it is necessary to investigate whether ASGR1 exerts additional effects on the progression of liver injuries. This includes exploring its potential role in mediating the regulation of hepatocytes on inflammatory cells and hepatic stellate cells during liver inflammation and fibrosis. Second, as liver biopsy is not a routine component of clinical care for patients with liver injury, our study focused on

**Fig. 6 | ASGR1 facilitates lysosomal degradation of GP73 in a clathrin-dependent manner. a** Immunofluorescence analysis of HepG2 cells incubated with fluorescent-labeled recombinant GP73 (red) at indicated time points. Scale bars, 5 μm, $n = 3$ biologically independent samples. **b** Immunofluorescence analysis of HepG2 cells with or without overexpression of *ASGR1* incubated with fluorescent-labeled recombinant GP73 (red) in the presence or absence of Asialofetuin A (30 μg/mL). Scale bars, 5 μm. **c** Quantification of the fluorescence intensity of GP73 shown in **b** ($n = 3$ per group). **d** Immunofluorescence analysis of HepG2 cells with or without overexpression of *ASGR1* transfected with si-*clathrin heavy chain* (si-*CHC*) or si-*control* and incubated with fluorescent-labeled recombinant GP73 (red). Scale bars, 5 μm. **e** Quantification of the fluorescence intensity of GP73 shown in **d** ($n = 3$ per group). **f** Immunofluorescence analysis of HepG2 cells with or without *ASGR1* overexpression incubated with fluorescent-labeled recombinant GP73 (red) and anti-LAMP1 antibody (green) in the presence or absence of chloroquine (5 μM). Scale bars, 5 μm. **g** Quantification of GP73-lysosomal-associated membrane protein 1 (GP73-LAMP1) colocalization shown in **f**. **h** Immunofluorescence analysis of *ASGR1*-overexpressing HepG2 cells with anti-LAMP1 antibody staining (green), as well as transfected with si-*CHC* or si-*control* after 6 h incubation with fluorescent-labeled recombinant GP73 (red) ($n = 3$ per group). Scale bars, 5 μm. **i** Quantification of GP73-LAMP1 colocalization shown in **h** ($n = 3$ per group). Data are presented as mean ± SEM. *P* values were calculated by one-way ANOVA with Dunnett's multiple comparisons test. **P* < 0.05. Source data are provided as a Source Data file.

samples obtained from patients with liver fibrosis or cirrhosis, and not directly from those with liver injury. This limitation emphasizes the need for further investigation of the role of ASGR1 in patients with liver injury. Third, in addition to GP73, there may be other targets involved in ASGR1-regulated liver injury. Studies exploring different ligands of ASGR1 are urgently needed and eagerly anticipated to provide a more comprehensive understanding of its molecular mechanisms. Lastly, although we observed liver damage in *Asgr1⁻/⁻* mice, a direct assessment of the correlation between liver health and *ASGR1* variants is essential.

In summary, this study provides evidence supporting the protective role of ASGR1 against liver injury. ASGR1 deficiency results in elevated circulating levels of GP73, subsequently triggering ER stress by enhancing the interaction between GP73 and BIP, ultimately leading to liver injury. Remarkably, neutralization of GP73 markedly mitigated this liver injury. Together, our findings reveal *ASGR1* as a candidate underlying genetic predisposition for liver injury, calling special attention to the potential risks associated with ASGR1 inhibition as a therapeutic strategy for the prevention and treatment of CAD.

## Methods

All human studies were approved by the Medical Ethics Committee of Xi-Jing Hospital of the Air Force Medical University (approval number: KY20172013-1 and KY20232280-X-1). The animal studies were approved by the Animal Ethical and Welfare Committee of Northwest A&F University and performed in accordance with all regulatory standards (approval number: NWAFU-314023743).

### Human samples
Liver biopsies and serum were collected from 6 patients with liver fibrosis, 10 patients with cirrhosis, 18 patients with hepatocellular carcinoma (stage I, $n = 6$; stage III, $n = 6$; stage III, $n = 6$), and 15 healthy living liver donors at the Xi-Jing Hospital of the Air Force Medical University. The patients' diagnosis, age, and sex were shown in Supplementary Table S1. Written informed consent was obtained from each participant.

### Animal studies
C57BL/6J mice were obtained from the animal center of Xi'an Jiao Tong University (Xi'an, China). ASGR1-deficient mice were created using CRISPR/Cas9 targeting system. All mice were housed in the animal facility at Northwest A&F University under standard conditions with free access to food and water. The light was on from 7am to 7 pm, with the temperature kept at 21–24 °C and humidity at 40–70%. Male mice were randomly divided into different groups as specified. Mice were euthanized by $CO_2$ inhalation following AVMA and institutional guidelines. Tissues were immediately snap-frozen and stored at −80 °C, or formalin-fixed for subsequent histological evaluation.

### Generation of ASGR1-deficient mice
*Asgr1⁻/⁻* mice were generated by CRISPR/Cas9-mediated gene targeting. Two single guide RNAs (sgRNAs) were designed to target exons 2 to 9 of the mouse Asgr1 gene using online tools (http:crispr.mit.edu/)

as illustrated in Supplementary Fig. S3. The sgRNAs and donors were co-injected into zygotes of C57BL/6J. Genomic DNA was extracted from tail biopsies for PCR genotyping and sequencing using Tiangen Bio-Universal Genomic DNA Extraction kit.

### AAV8-mediated *Asgr1* overexpression
The AAV8 delivery system that overexpresses *Asgr1* gene in mouse livers was constructed by Hanbio Tech (Shanghai, China), with AAV-NC as control. Mice were injected with a single dose of virus (100 μl) containing $2 × 10^{11}$ AAV8 vector genomes via tail vein for 4 weeks.

### Mouse models of chronic liver injury induced by CCl4
WT, *Asgr1⁻/⁻* or AAV-*Asgr1* mice were intraperitoneally injected with CCl4 (10% in olive oil, 1 ml/kg, twice a week) or vehicle (1 ml/kg of olive oil) for 6 weeks[66].

### Mouse models of acute liver injury induced by APAP
WT, *Asgr1⁻/⁻* or AAV-*Asgr1* mice were fasted overnight and then intraperitoneally injected with a single dose of 400 mg/kg APAP or vehicle (PBS)[67].

### Treatment of mice with TUDCA
Eight-week-old *Asgr1⁻/⁻* mice and WT controls were intraperitoneally injected with CCl4 (1 ml/kg body weight, twice a week for 6 weeks). During the last 4 weeks, mice received either an ER stress inhibitor TUDCA (500 mg/kg body weight, every two days) or vehicle via intraperitoneal injection. For APAP intoxication, eight-week-old *Asgr1⁻/⁻* and WT controls treated with a single dose of APAP (400 mg/kg body weight) were intraperitoneally injected with TUDCA (250 mg/kg body weight) or vehicle[3].

### GP73 neutralization
Eight-week-old *Asgr1⁻/⁻* and WT mice were intraperitoneally injected with CCl4 (1 ml/kg body weight, twice a week for 6 weeks) and during the last 4 weeks mice were intraperitoneally injected with IgG or anti-GP73 (10 mg/kg body weight, twice a week for 4 weeks, generated by Hotgen Biotech Co., Ltd.). For APAP intoxication, eight-week-old *Asgr1⁻/⁻* and WT mice were intraperitoneally injected with anti-GP73 (10 mg/kg body weight) or IgG 10 h after APAP injection (400 mg/kg body weight)[31].

### Treatment of mice with Tunicamycin (Tm)
Eight-week-old *Asgr1⁻/⁻* and WT mice were intraperitoneally injected with anti-GP73 (10 mg/kg body weight) along with Tm (2 mg/kg body weight, Beyotime Biotech Inc) or vehicle 10 h after APAP injection (400 mg/kg body weight)[31].

### Survival study
Eight-week-old *Asgr1⁻/⁻* and WT mice were intraperitoneally injected with anti-GP73 (10 mg/kg body weight) or IgG at 10 h after a lethal dose of APAP challenge (650 mg/kg body weight). Mice were followed for 8 days to monitor survival[68].

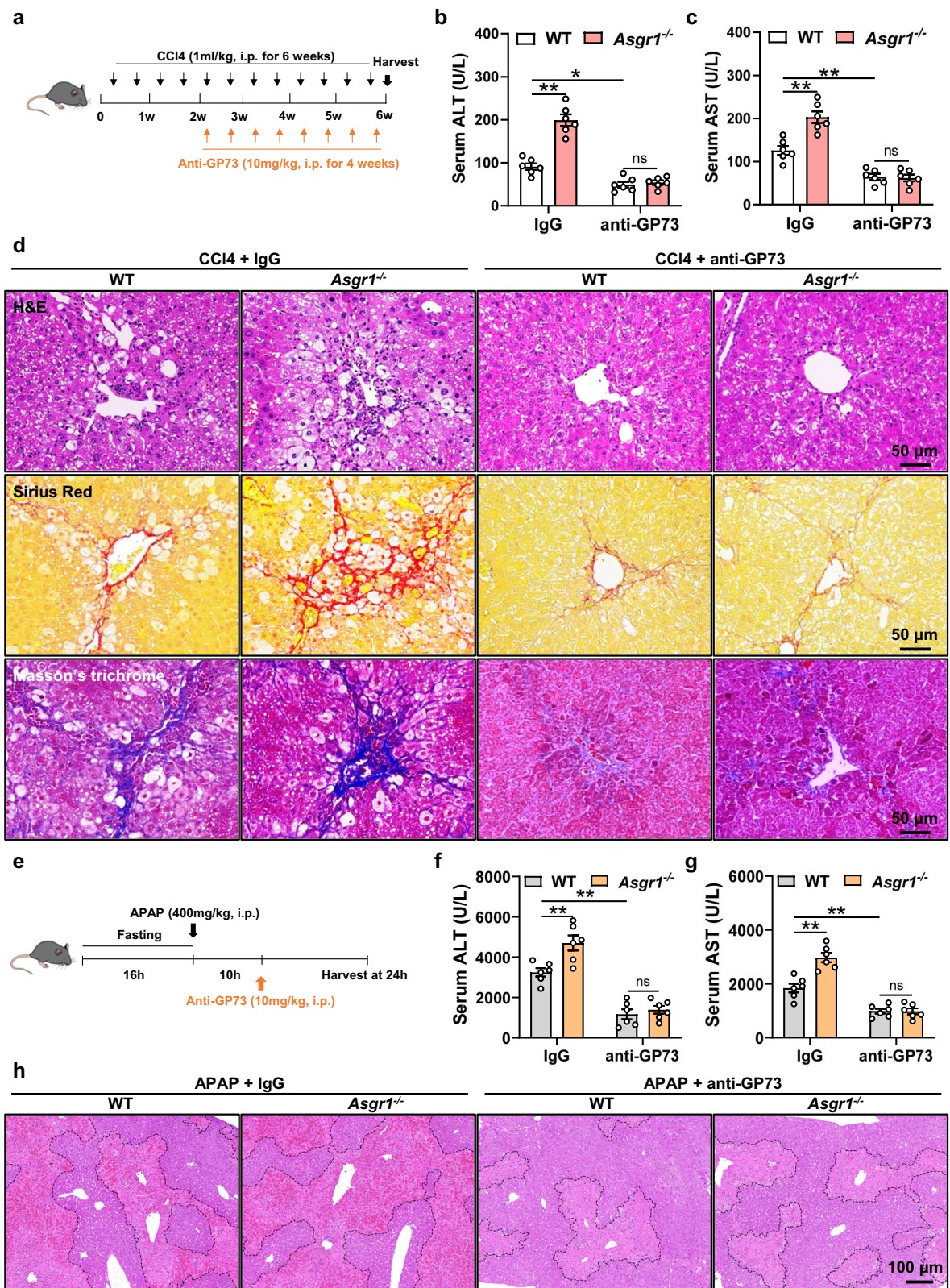

## Cell culture and treatment

HepG2 cells were obtained from the American Type Culture Collection (ATCC). Cells were cultured in Dulbecco's Modified Eagle's Medium (DMEM) supplemented with 10% FBS (ZETA LIFE, California, USA) and 10 U/ml penicillin/streptomycin at 37 °C in 5% $CO_2$. To detect GP73 endocytosis, cells with ASGR1-OE or knockdown were treated with fluorescent-labeled recombinant GP73 (prepared using Alexa Fluor 647 labeling kit). For the detection of GP73 degradation, cells with ASGR1 overexpression were treated with chloroquine (5 μM) or 30 μg/ml asialofetuin A (prepared as previously described[20]).

## siRNA knockdown

Gene-specific siRNA was designed and synthesized by GenePharma (Shanghai, China). siRNA was transfected individually into cells using

**Fig. 7 | Neutralization of GP73 attenuates ASGR1 deficiency-induced liver injury. a** Schematic diagram of mice treatment. 8-week-old *Asgr1⁻ᐟ⁻* and WT mice were intraperitoneally injected with CCl4 (1 ml/kg body weight, twice a week for 6 weeks). During the last 4 weeks, mice were received either anti-GP73 (10 mg/kg body weight, twice a week for 4 weeks) or IgG. **b** Serum levels of ALT (*n* = 6 per group; CCl4+IgG+WT *vs*. CCl4+IgG+*Asgr1⁻ᐟ⁻*, *P* = 0.00000018; CCl4+IgG+WT *vs*. CCl4+anti-GP73 + WT, *P* = 0.0148; CCl4+anti-GP73 + WT *vs*. CCl4+anti-GP73+*Asgr1⁻ᐟ⁻*, *P* = 0.9798). **c** Serum levels of AST (*n* = 6 per group; CCl4+IgG+WT *vs*. CCl4+IgG+*Asgr1⁻ᐟ⁻*, *P* = 0.0001; CCl4+IgG+WT *vs*. CCl4+anti-GP73 + WT, *P* = 0.0015; CCl4+anti-GP73 + WT *vs*. CCl4+anti-GP73+*Asgr1⁻ᐟ⁻*, *P* = 0.9969). **d** H&E staining, Sirius red staining and Masson's trichrome staining of liver sections. Scale bars, 50 µm. **e** Schematic diagram of mice treatment. 8-week-old *Asgr1⁻ᐟ⁻*

and WT mice were intraperitoneally injected with anti-GP73 (10 mg/kg body weight) or IgG at 10 hours after APAP injection (400 mg/kg body weight). **f** Serum levels of ALT (*n* = 6 per group; APAP+IgG+WT *vs*. APAP+IgG+*Asgr1⁻ᐟ⁻*, *P* = 0.0047; APAP+IgG+WT *vs*. APAP+anti-GP73 + WT, *P* = 0.0001; APAP+anti-GP73 + WT *vs*. APAP+anti-GP73+*Asgr1⁻ᐟ⁻*, *P* = 0.9337). **g** Serum levels of AST (*n* = 6 per group; APAP+IgG+WT *vs*. APAP+IgG+*Asgr1⁻ᐟ⁻*, *P* = 0.000068; APAP+IgG+WT *vs*. APAP +anti-GP73 + WT, *P* = 0.0016; APAP+anti-GP73 + WT *vs*. APAP+anti-GP73+*Asgr1⁻ᐟ⁻*, *P* = 0.9999). **h** H&E staining of liver sections. Scale bars, 100 µm. Necrotic areas were encircled. Data are presented as mean ± SEM. *P* values were calculated by two-way ANOVA with Tukey's multiple comparison test. \**P* < 0.05, \*\**P* < 0.01. Source data are provided as a Source Data file.

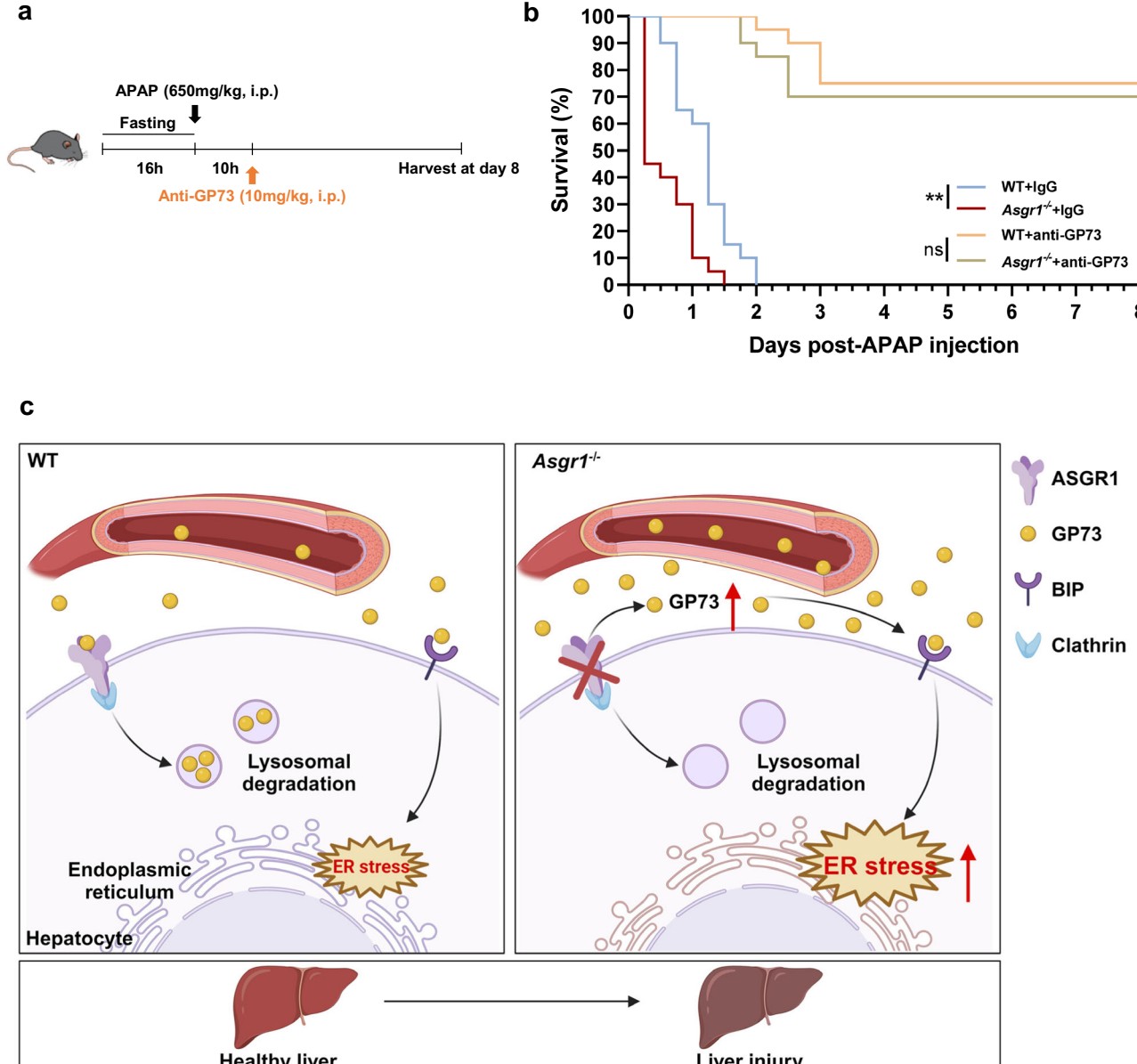

**Fig. 8 | GP73 neutralization improves survival in ASGR1-deficient mice treated with APAP. a** Schematic diagram of mice treatment. 8-week-old *Asgr1⁻ᐟ⁻* and WT mice were intraperitoneally injected with anti-GP73 (10 mg/kg body weight) or IgG 10 hours after a lethal dose of APAP injection (650 mg/kg body weight). **b** Survival curves of mice treated with either IgG or anti-GP73 10 h after lethal APAP dosing (*n* = 20; WT+IgG *vs*. *Asgr1⁻ᐟ⁻*+IgG, *P* = 0.00034; WT+anti-GP73 *vs*. *Asgr1⁻ᐟ⁻*+anti-GP73, *P* = 0.5957). Data are presented as mean ± SEM. *P* values were calculated by Log-Rank test. \*\**P* < 0.01. **c** Working model of ASGR1 in liver injury. Left, in the presence

of ASGR1, it binds to and mediates GP73 endocytosis and lysosomal degradation to maintain the homeostasis of circulating GP73 levels. Right, ASGR1 deficiency inhibits GP73 endocytosis and degradation, leading to excessive accumulation of GP73 in the circulation. As a result, the interaction between accumulated GP73 and BIP is enhanced, leading to increased ER stress and liver injury. Working model of ASGR1 in liver injury created with BioRender.com. Source data are provided as a Source Data file.

Lipofectamine RNAiMAX transfection reagent following the manufacturer's instructions. The siRNA sequences were as follows: *CHC* siRNA, 5′-CCGGAAAUUUGAUGUCAAUACUUCA-3′; *ASGR1* siRNA, 5′-GCUGCUUGUGGUUGUCUGUGUTT-3′; Negative-control siRNA, 5′-UUCUCCGAACGUGUCACGUGUTT-3′.

## Plasmid construction and transfection

The HA-tag was added at the N-terminus of ASGR1 (WT), ASGR1 (truncation mutant), and ASGR1(3A) (alanine mutant) and then cloned into the pcDNA3.1-HA vector. The Flag-tag was added at the N-terminus of GP73 (WT), GP73 (truncation mutant), GP73-N109A, GP73-N144A and GP73-N398A (alanine mutants) and then cloned into the pcDNA3.1-Flag vector. The His-tag was added at the N-terminus of BIP and then cloned into the pet-28a vector. The plasmid was extracted by EndoFree Plasmid Midi Kit (CWBIO, China). HepG2 cells were transfected with the plasmids using Lipofectamine 3000 (L3000001, Invitrogen, California, USA).

## H&E staining, Sirius red staining, and Masson's trichrome staining

Liver specimens were fixed in 10% neutral buffered formalin overnight at 4 °C. Fixed tissues were processed through a series of solvents starting from ethanol solutions to xylenes. Specimens were then embedded in paraffin and cut into 4 μm sections. Sections were deparaffinized and hydrated, then stained with H&E, Sirius red, or Masson's trichrome by standard methods. The stained sections were visualized using an optical microscope camera (Olympus, Japan).

## Quantitative real-time PCR

Total RNA was isolated from mouse liver using TRIzol reagent (Tsingke Biotech, Xian, China). 2 μg total RNA was then used for reverse transcription reaction using the cDNA synthesis kit (Deeyee, China). SYBR Green (Accurate Biotechnology, Hunan, Changsha, China) was used for quantitative real-time PCR. Each sample was analyzed with GAPDH as the internal control. The quantification of mRNA expression was calculated using the ($2^{-\Delta\Delta Ct}$) method. The primer pairs used in this study were listed in Supplementary Table S2.

## Western blot

Cells and liver tissues were homogenized in RIPA buffer containing a cocktail of protease inhibitors. Proteins in lysates were resolved by SDS-PAGE and immunoblotted with the indicated primary antibodies and their corresponding HRP-conjugated secondary antibodies. Blots were developed with chemiluminescent HRP substrate and imaged. Antibodies used for western blotting were listed in Supplementary Table S3.

## Immunofluorescent (IF) staining

IF staining was performed using liver sections as previously described[69]. Livers were paraffin-embedded. Then, the paraffin sections were deparaffinized and rehydrated, followed by antigen retrieval using sodium citrate buffer. The sections were then incubated in wash buffer and blocked for 2 h at room temperature (RT). Primary antibodies were applied to the sections at standardized concentrations and incubated overnight at 4 °C. Next, fluorescent antibodies were applied for 1 h at RT. Nuclei in all images were stained with DAPI (Solarbio, Beijing). All antibodies used and their respective dilutions were listed in Supplementary Table S3.

## Measurement of serum biochemical indicators and hepatic hydroxyproline

Serum levels of ALT, AST, ALP, and GGT were measured using commercially available assay kits (Jiancheng, Nanjing) according to the manufacturer's instructions. Serum levels of GP73 were measured using the ELISA kit (mlbio, Shanghai) following the manufacturer's instructions. Hepatic content of hydroxyproline was determined using a hydroxyproline colorimetric assay kit according to the manufacturer's instructions.

## Immunoprecipitation/mass spectrometry

Immunoprecipitation (IP) assays were performed as previously described[31]. HepG2 cells were cotransfected with the indicated plasmid for 24 h. Cell lysates were prepared by sonication in NP40 buffer (1% NP40, 150 mM NaCl and 40 mM Tris pH 7.5) for immunoprecipitation. Proteins were immunoprecipitated using anti-HA magnetic beads (MedChemExpress, USA) and anti-Flag magnetic beads (Bimake, China) at 4 °C overnight. The obtained beads were washed and boiled in 2×loading buffer at 95 °C for 10 min. The immune complex was collected and subjected to western blotting analysis with the corresponding primary antibodies and secondary antibodies. For mass spectrometry, immunoprecipitated ASGR1 was eluted using HA peptide (Anaspec), and the eluted proteins were precipitated with trichloroacetic acid and digested with trypsin. The resulting tryptic peptides were desalted over C18 resin and then loaded onto an LTQ linear ion trap mass spectrometer (Thermo Finnigan) for LC-MS/MS analyses. MS/MS spectra were searched using SEQUEST against a target-decoy database of tryptic peptides, and candidate proteins were screened with an online tool (https://www.uniprot.org/).

## RNA-seq and Gene set enrichment analysis (GSEA)

Liver samples from *Asgr1−/−* and WT mice were used for RNA-seq analyses. Analyses were performed by Majorbio, using the Illumina noveseq6000 platform. The RNA-seq data were aligned to corresponding reference genomes (mm10) using HiSat2 and TopHat2. The gene expression level was then estimated as transcripts per million (TPM), and the differentially expressed genes were defined with *P* adjust <0.05 and a fold change of ±2 or more using DESeq2. Statistically enriched gene pathways were analyzed by GSEA using clusterProfiler and enrichplot. All the sequencing data have been submitted to the National Center for Biotechnology Information's Gene Expression Omnibus data bank (accession number: GSE232677).

## Statistical analyses

Data were presented as mean ± standard error of the mean (SEM). Statistical comparisons between two groups were analyzed by Student's *t*-test and for more than 2 groups, by one-way ANOVA analysis followed by multiple comparisons correction using Dunnett (when several experimental groups were compared to a single control group) or Tukey (when several conditions were compared to each other within one experiment) or two-way ANOVA analysis followed by multiple comparisons correction using Tukey. Survival was measured by the Kaplan-Meier method and analyzed by the Log-rank (Mantel-Cox) test. Statistical correlation was analyzed using Pearson's Correlation Coefficient (two-tailed, confidence interval (CI) = 95%). Statistical analysis was performed using GraphPad Prism version 8.0.2 for Windows (GraphPad Software, USA, www.graphpad.com). *P* < 0.05 was considered significant.

## Reporting summary

Further information on research design is available in the Nature Portfolio Reporting Summary linked to this article.

# Data availability

All data are available in the main text or the supplementary materials. Source data are provided with this paper. All the sequencing data generated in this study have been deposited in the NCBI's Gene Expression Omnibus data bank under accession code GSE232677. Any additional information is available upon request to the corresponding author (Jiang Wei Wu, wujiangwei@nwafu.edu.cn). Source data are provided with this paper.

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

## Acknowledgements

This research was supported by: National Natural Science Foundation (32370569 and 32070602) (J.W.); National Key Research and Development Program of China (2021YFF1000602) (J.W.); the Program for Shaanxi Science & Technology (2023-CX-TD-57) (J.W.). We thank the Life Science Research Core Services (LSRCS), Northwest A&F University, for providing the confocal microscope (Xiaorui Liu) and mass spectrometer (Meijuan Ren). Working model of ASGR1 in liver injury created with BioRender.com. Acknowledging the contribution of the drawings' respective authors and of Scidraw.

## Author contributions

Z.Z. and J-W.W. conceptualization; Z.Z., X-K.L., X.Z., Z-W.S., M.J. methodology; Y-Y.Z., J-Y.X., J-F.L., Y-N.J. investigation; Z.Z., Z-W.S., K.L., Q.G., C-Q.X., X-G.Z., K-S.T. data curation; Z.Z., X-K.L., Y-Y.Z. Writing-Original Draft; B.X., J-W.W. Writing-Review & Editing.

## Competing interests

The authors declare no competing interests.
