## [Peer Review File · Nature Communications]

REVIEWER COMMENTS

Reviewer #1 (Remarks to the Author):

In this manuscript, Zhang et al studied the role of ASGR1 in liver damage. Using mouse models of acute and chronic liver injury, they show that ASGR1 plays a protective role by interfering with the ER stress mediator GP73. The study is interesting and identified a potential therapeutic target for liver diseases.

Major comments

1. To study the role of GP73 on liver injury and survival, the authors challenged mice with a toxic (Fig. 7E-G) or a lethal (Fig. 8A-B) dose of APAP, and 10 hours later administered anti-GP73 neutralizing antibody:

- It is well known that liver injury occurs relatively fast in mice after APAP intoxication with a peak of oxidative stress at 2 hours and a maximum damage at 8-10 hours (doi: 10.1016/j.jhep.2022.01.020). Therefore, at the time when the GP73 neutralizing antibody was administered, the destruction process following APAP overdose might already be over and the regeneration phase might be dominating. How can the authors explain the observed protection against APAP following the late administration of anti-GP73? Does it enhance regeneration?

- Also, the data of the liver enzymes (Fig. 7F-G) are not consistent with the histopathology data shown in Fig. 7H; while the enzyme activities are strongly ameliorated by the neutralizing antibody, the HE staining shows massive pericentral hepatocyte necrosis in the same group; please check and clarify.

2. Since both APAP and CCl4 require metabolic activation by the same enzyme (CYP2E1), the influence of ASGR1 interventions (deficiency or overexpression) on CYP2E1 expression should be added as a control experiment.

Minor comments

- The image of the PBS control group in Fig. 1D shows a portal vein and periportal hepatocytes, which is not comparable to the APAP group which shows central veins and pericentral hepatocytes; please replace with comparable images.

- The entire manuscript should be checked for grammar/typing mistakes; for example, in the limitations of the study section, what are the 'satellite' cells?

Reviewer #2 (Remarks to the Author):

In this study, the authors suggested a protective effect of ASGR1 against liver injury, and ASGR1 deficiency increases circulating levels of GP73 and subsequently activates ER stress via elevating the interaction between GP73 and BIP, and consequently, leading to liver injury. Moreover, their data also demonstrated that neutralization of GP73 markedly attenuated the liver injury, in mouse model.

Though the work is interesting, there are some concerns need to be addressed:

1. Fibrosis is an important stage prior to cirrhosis, how about ASGR1's expression in fibrotic hepatocytes?
2. The link of GP73 to liver disease was firstly reported by the finding that is highly expressed in liver cancer. How about GP73's relationship with ASGR1 in the malignant transformed hepatic cells.
3. The authors showed that ASGR1 deficiency increases circulating levels of GP73, and ASGR1 physically binding to GP73 protein, would ASGR1 somehow involved in the digestion and release of GP73 into circulation?
4. Last but important question: the authors reported in the submission that neutralization of GP73 markedly attenuated the liver injury, and GP73 intercellular interacted with BIP to active ER stress. How to explain such outer an inter difference?

Reviewer #3 (Remarks to the Author):

Manuscript Number: NAR-01594-Y-2023

Manuscript Title: Deficiency 1 of ASGR1 promotes liver injury by increasing
2 GP73-mediated hepatic endoplasmic reticulum stress

Genetic factors leading to the initiation and progression of the liver injury remain unknown and it is critical for developing therapeutic intervention. In this report authors evaluated the role of asialoglycoprotein receptor (ASGR1) in liver injury of mice. Authors demonstrate that ASGR1 deficiency exacerbates the liver injury. In addition, they also demonstrate that ASGR1 overexpression improves APAP-induced acute and CCl4-induced chronic liver injuries in mice. Their studied to understand mechanism reveals that ASGR1 binds GP73 which in turn trigger lysosomal degradation and ER stress play a critical role in liver injury of mice. Authors also found that cirrhotic patients with decreased ASGR1 expression display increased serum GP73 levels and hepatic stress. The finding reported in this manuscript identify ASGR1 as a novel candidate underlying genetic cause to liver injury. This article is well written, and all experiments were executed diligently. This article is suitable for publication in Nature Communications.

Point-by-point response to the reviewers' comments

Reviewer #1 (Remarks to the Author):

In this manuscript, Zhang et al studied the role of ASGR1 in liver damage. Using mouse models of acute and chronic liver injury, they show that ASGR1 plays a protective role by interfering with the ER stress mediator GP73. The study is interesting and identified a potential therapeutic target for liver diseases.

Reply: We highly appreciate your careful and insightful review of our manuscript.

Major comments

1. To study the role of GP73 on liver injury and survival, the authors challenged mice with a toxic (Fig. 7E-G) or a lethal (Fig. 8A-B) dose of APAP, and 10 hours later administered anti-GP73 neutralizing antibody:

- It is well known that liver injury occurs relatively fast in mice after APAP intoxication with a peak of oxidative stress at 2 hours and a maximum damage at 8-10 hours (doi: 10.1016/j.jhep.2022.01.020). Therefore, at the time when the GP73 neutralizing antibody was administered, the destruction process following APAP overdose might already be over and the regeneration phase might be dominating. How can the authors explain the observed protection against APAP following the late administration of anti-GP73? Does it enhance regeneration?

Reply: Thanks for pointing out the relevant reference in the field. We carefully read this paper (Ghallab, et al. *J Hepatol*, PMID: 35131407) and other related publications (Chen, et al. *J Hepatol*, PMID: 36368597). As you emphasized here, it is well accepted that liver injury occurs relatively fast in mice after APAP intoxication with a peak of oxidative stress at 2 hours and a maximum damage at 8-10 hours. In this study, we also observed elevated oxidative stress in livers of both WT and ASGR1-deficient mice at 2h after APAP treatment as evidenced by decreased levels of GSH, an important cellular antioxidant (**Fig. R1**). GSH levels was recovered at 10 hours after APAP intoxication (**Fig. R1**). Nonetheless, hepatic GSH levels were not significantly different between WT and ASGR1-deficient mice at either 2 or 10 hours after APAP intoxication (**Fig. R1**). Thus, it is unlikely that oxidative stress contributes to ASGR1 deficiency induced liver injury in mice treated with APAP.

Fig. R1. ASGR1 deficiency does not affect hepatic oxidative stress in mice treated with APAP. GSH concentrations in livers of WT or *Asgr1*^{-/-} mice treated with APAP (n = 6). Data are presented as mean ± SEM. *P* values were calculated by two-tailed unpaired t-test. **P* < 0.05, ***P* < 0.01.

Besides oxidative stress, accumulating evidence suggests that hepatic ER stress plays a crucial role in APAP-induced liver injury (Torres, et al. *Gastroenterology*, PMID: 31029706; Li, et al. *Cell Death Differ*, PMID:31827236; Uzi, et al. *J Hepatol*, PMID: 23665281). Unlike oxidative stress, which peaks at 2 hours after APAP intoxication, protein levels of ER stress marker CHOP were gradually elevated from 2 to 16 hours in livers of WT mice after APAP exposure (Li, et al. *Cell Death Differ*, PMID:31827236). Conversely, pharmacological inhibition of ER stress markedly reduced APAP-induced liver injury (Torres, et al. *Gastroenterology*, PMID: 31029706). Here, we showed higher levels of hepatic ER stress in ASGR1-deficient mice than in their respective WT controls at either 2 or 10 h after APAP treatment, as evidenced by increased hepatic mRNA and protein levels of ER stress markers BIP and CHOP (**Fig. R2a-c**). Longitudinally, hepatic ER stress was higher in mice after 10h APAP treatment than in those of 2h (**Fig. R2a-c**). Together, several types of stress collectively contribute to APAP-induced liver injury: oxidative stress occurs in the early stage and reaches peak levels at 2 hours after APAP treatment, whereas ER stress is activated from 2h and lasts for more than 10 hours after APAP treatment. The latter is associated with ASGR1-induced liver injury.

Fig. R2. ASGR1 deficiency aggravates hepatic ER stress in mice treated with APAP. (a to c) Relative hepatic mRNA and protein expression of BIP and CHOP in livers of WT or *Asgr1*^{-/-} mice treated with APAP. Data are presented as mean \pm SEM. *P* values were calculated by two-tailed unpaired t-test. **P* < 0.05, ***P* < 0.01.

We completely agree with you that maximum liver damage occurs after 8-10 hours APAP challenge. The reason we chose 10 hours after APAP treatment to test the therapeutic potential of GP73 neutralizing antibody for liver injury and survival is that (i) APAP overdose patients often present late in the hospital due to initially no immediate sedative effect and few symptoms until abdominal pain and nausea develop between 12 and 24 hours (Lee, et al. *J Hepatol*, PMID: 28734939). (ii) The only FDA-approved standard antidote for APAP intoxication NAC is effective in the early stages, while its efficacy is limited to the therapeutic window (within 8 hours) (Du, Kuo et al. *Redox Biol*, PMID: 27744120; Ghallab, A., et al. *J Hepatol*, PMID: 35131407). Therefore, exploring therapeutic agents for late stage APAP intoxication is urgently needed. Great efforts have been devoted to this direction: Widjaja, et al. tested a neutralizing antibody against IL11RA 10 h after APAP (400 mg kg⁻¹) challenge in mice and found significantly alleviated liver damage (Widjaja, et al. *Sci Transl Med*, PMID: 34108253); Lewis, et al. found that alternatively activated macrophages (AAM) supplementation 16 hours after APAP treatment significantly alleviated liver injury in mice (Lewis, et al. *J Hepatol*, PMID: 32169610). These findings are quite promising and prompted us to test whether GP73 neutralizing antibody exerts similar ameliorating effects at this stage (late stage). Thus, we chose to administer GP73 neutralizing antibody 10h after APAP intoxication.

As you predicted, here we showed that serum levels of biomarkers of liver function ALT and AST were significantly increased at 10h after APAP intoxication in both WT and ASGR1-deficient mice (**Fig. R3a, b**). H&E staining of liver sections also revealed increased liver injury at this time point (10h) (**Fig. R3c**). Hepatic mRNA levels of *cyclin A2/B1/D1/E1* and the number of Ki67 positive cells were significantly increased at 10h after APAP intoxication, indicative of liver regeneration (**Fig. R3d-h**). Compared with WT mice treated with APAP, ASGR1-deficient mice showed more severe liver injury and less liver regeneration at 10h after APAP treatment (**Fig. R3d-h**). Neutralization of GP73 alleviated liver injury as evidenced by decreased levels of ALT and AST, as well as reduced hepatocyte necrosis (**Fig. 7e-h**). Meanwhile, GP73 neutralization also up-regulated mRNA levels of *cyclin A2/B1/D1/E1* and increased the number of Ki67 positive cells in the liver, indicating that it promotes liver regeneration (**Fig. S18a-e**). The differences in the degree of liver injury and regeneration between WT and ASGR1-deficient mice disappeared after GP73 neutralization (**Fig. 7e-h, Fig. S18a-e**).

Fig. R3. ASGR1 deficiency increases liver injury and decreases liver regeneration in mice with APAP treatment. 8-week-old *Asgr1*^{-/-} and WT mice were intraperitoneally injected with PBS or APAP (400mg/kg body weight) (n=6). **(a, b)** Serum levels of ALT and AST in mice after 10 hours of treatment. **(c)** H&E staining of liver sections. Scale bars, 100 μm. Necrotic areas were encircled. **(d to g)** Relative hepatic mRNA expression of *cyclin A2/B1/D1/E1*. **(h)** Representative images of Ki-67 immunofluorescent staining. Scale bars, 50 μm. Data are presented as mean ± SEM. *P* values were calculated by two-way ANOVA with Tukey's multiple comparison test. **P* < 0.05, ***P* < 0.01.

Fig. 7. (e) Schematic diagram of mice treatment. 8-week-old *Asgr1*^{-/-} and WT mice were intraperitoneally injected with anti-GP73 (10mg/kg body weight) or IgG 10 hours after APAP injection (400mg/kg body weight). (f and g) Serum levels of ALT and AST. (h) H&E staining of liver sections. Scale bars, 100 μ m. Necrotic areas were encircled. Data are presented as mean \pm SEM. *P* values were calculated by two-way ANOVA with Tukey's multiple comparison test. **P* < 0.05, ***P* < 0.01.

Fig. S18. GP73 neutralization rescues ASGR1 deficiency-induced impaired liver regeneration in APAP-treated mice. 8-week-old *Asgr1*^{-/-} and WT mice were intraperitoneally injected with anti-GP73 (10mg/kg body weight) or IgG 10 hours after APAP injection (400mg/kg body weight) (n=6). **(a to d)** Relative hepatic mRNA expression of *cyclin A2/B1/D1/E1*. **(e)** Representative images of Ki-67 immunofluorescent staining. Scale bars, 50 μ m. Data are presented as mean \pm SEM. *P* values were calculated by two-way ANOVA with Tukey's multiple comparison test. **P* < 0.05, ***P* < 0.01.

To further explore mechanism underlying the observed protection against APAP following the administration of anti-GP73 in ASGR1-deficient mice, we examined hepatic ER stress based on the fact that (i) ASGR1 deficiency significantly increased ER stress (**Fig. 4, Fig. R2**); (2) pharmacological inhibition of ER stress by TUDCA significantly reduced ASGR1 deficiency induced liver injury (**Fig. 4, Fig. S10**). Our results showed that the mRNA and protein levels of ER stress markers BIP and CHOP were increased in livers of ASGR1-deficient mice after APAP intoxication (**Fig. S17**), whereas GP73 neutralization significantly decreased their levels (**Fig. S17**), suggesting that the beneficial effect of GP73 neutralization on APAP-induced liver injury in ASGR1-deficient mice might be attributable to the reduction of ER stress. It is well established that ER stress marker (CHOP) plays a pro-damage role in response to APAP intoxication, in part by curtailing the regeneration phase following liver damage (Uzi, et al. *J Hepatol*, PMID: 23665281). Since we observed reduced ER stress and increased liver regeneration upon GP73 neutralization, we hypothesize that neutralization of GP73 may promote liver regeneration by inhibiting ER stress. To test it, ASGR1-deficient mice and WT controls with GP73 neutralization were further treated with an ER stress agonist tunicamycin (Tm). This treatment dramatically inhibited liver regeneration shown in anti-GP73 treated WT or *Asgr1*^{-/-} mice (**Fig. S19**), suggesting that ER stress mediates the effect of GP73 neutralization on liver regeneration in mice.

Fig. S10. Pharmacological inhibition of ER stress attenuates ASGR1 deficiency-induced liver injury in APAP-treated mice. (a) Schematic diagram of mice treatment. 8-week-old *Asgr1*^{-/-} and WT mice with APAP intoxication (400mg/kg body weight) were treated with TUDCA (250mg/kg body weight) or vehicle (n=6). (b-c) Serum levels of ALT and AST. (d) H&E staining of liver sections. Scale bars, 100 μ m. Necrotic areas were encircled. Data are presented as mean \pm SEM. *P* values were calculated by two-way ANOVA with Tukey's multiple comparison test. **P* < 0.05, ***P* < 0.01.

Fig. S17. GP73 neutralization attenuates ASGR1 deficiency-induced ER stress in APAP-treated mice. 8-week-old *Asgr1*^{-/-} and WT mice were intraperitoneally injected with anti-GP73 (10mg/kg body weight) or IgG 10 hours after APAP injection (400mg/kg body weight) (n=6). **(a to c)** Relative hepatic mRNA and protein expression of BIP and CHOP. **(d and e)** Quantification of BIP and CHOP protein levels (n=3). Data are presented as mean \pm SEM. *P* values were calculated by two-way ANOVA with Tukey's multiple comparison test. **P* < 0.05, ***P* < 0.01.

Fig. S19. ER stress agonist inhibits anti-GP73 induced liver regeneration. 8-week-old *Asgr1*^{-/-} and WT mice intraperitoneally injected with anti-GP73 (10mg/kg body weight) were treated with Tm (2mg/kg body weight) or vehicle 10 hours after APAP injection (400mg/kg body weight) (n=6). **(a to d)** Relative hepatic mRNA expression of *cyclin A2/B1/D1/E1*. **(e)** Representative images of Ki-67 immunofluorescent staining. Scale bars, 50 μ m. Data are presented as mean \pm SEM. *P* values were calculated by two-way ANOVA with Tukey's multiple comparison test. **P* < 0.05, ***P* < 0.01.

For your comment that “the data of the liver enzymes (Fig. 7F-G) are not consistent with the histopathology data shown in Fig. 7H; while the enzyme activities are strongly ameliorated by the neutralizing antibody, the HE staining shows massive pericentral hepatocyte necrosis in the same group; please check and clarify.”

Reply: Thanks for your careful examination of the seemingly contradictory results between ameliorated liver transaminase levels and massive hepatocyte necrosis after GP73 neutralization. We would like to emphasize that although levels of liver enzymes are strongly ameliorated by GP73 neutralization in mice treated with APAP as shown in **Fig. 7f-g**, they remained at higher levels than those in non-APAP treated normal mice which are normally below 100U/L. Although anti-GP73 treated mice still had massive pericentral hepatocyte necrosis, the extent of necrosis was less severe than that of IgG-treated WT and ASGR1-deficient mice (**Fig. 7h, Fig, R4**). Hence, our results demonstrate that GP73 neutralization significantly alleviates, but not enough to completely normalize, APAP-induced liver injury in mice. These results are consistent with those of previous studies, which showed either IL11RA neutralization after 10 hours of APAP (Widjaja, et al. *Sci Transl Med*, PMID: 34108253) or alternatively activated macrophages supplementation after 16 hours of APAP treatment (Lewis, et al. *J Hepatol*, PMID: 32169610) significantly reduced levels of ALT and AST, but not completely restore liver injury and still showed hepatocyte necrosis.

For late stage APAP intoxication as we tested here (10h), amelioration of liver injury and avoidance of life threatening medical conditions are crucial, although complete normalization and recovery is the ultimate goal. The effect of GP73 neutralization may have a better effect on early stage APAP-induced liver injury. However, this is not the main aim of our study since early stage APAP intoxication can be well treated by the current medicine NAC. Future studies are necessary to test the combined effects of GP73 neutralization and other agents on APAP-induced late stage liver injury.

Fig. R4. Biological duplication of H&E staining of liver sections in Figure 7h. Scale bars, 100 μm . Necrotic areas were encircled.

2. Since both APAP and CCl4 require metabolic activation by the same enzyme (CYP2E1), the influence of ASGR1 interventions (deficiency or overexpression) on CYP2E1 expression should be added as a control experiment.

Reply: Thanks for your suggestion. The cytochrome P450 enzyme CYP2E1 mediates metabolism of APAP and CCl4 resulting in the formation of reactive metabolites that cause deadly hepatotoxicity (Lee, et al. *J Biol Chem*, PMID: 8662637). As requested, we now measured the effect of ASGR1 on expression levels of CYP2E1 as a control experiment. ASGR1 deficiency did not affect the expression of CYP2E1 in APAP- or CCl4-treated mice, since their mRNA and protein levels were comparable between *Asgr1*^{-/-} mice and WT controls (**Fig. S6a-f**). Similarly, hepatic *Asgr1*-overexpressing mice exhibited similar CYP2E1 expression to their controls in APAP or CCl4 treated conditions (**Fig. R5a-f**). These results indicate that, despite playing an essential role in APAP or CCl4 induced liver injury, CYP2E1 expression is not affected by ASGR1.

Fig. S6. ASGR1 deficiency does not affect CYP2E1 expression in liver injured mice. (a and b) Relative hepatic mRNA and protein expression of CYP2E1 in WT or *Asgr1*^{-/-} mice treated with APAP. **(c)** Quantification of CYP2E1 protein levels (n=3). **(d and e)** Relative hepatic mRNA and protein expression of CYP2E1 in WT or *Asgr1*^{-/-} mice treated with CCl4. **(f)** Quantification of CYP2E1 protein levels (n=3). Data are presented as mean ± SEM. *P* values were calculated by two-tailed unpaired t-test. **P* < 0.05, ***P* < 0.01.

Fig. R5. ASGR1 overexpression does not affect CYP2E1 expression in liver injured mice. (a and b) Relative hepatic mRNA and protein expression of CYP2E1 in AAV-NC or AAV-Asgr1 mice treated with APAP. **(c)** Quantification of CYP2E1 protein levels (n=3). **(d and e)** Relative hepatic mRNA and protein expression of CYP2E1 in AAV-NC or AAV-Asgr1 mice treated with CCl4. **(f)** Quantification of CYP2E1 protein levels (n=3). Data are presented as mean \pm SEM. *P* values were calculated by two-tailed unpaired t-test. **P* < 0.05, ***P* < 0.01.

Minor comments

- The image of the PBS control group in Fig. 1D shows a portal vein and periportal hepatocytes, which is not comparable to the APAP group which shows central veins and pericentral hepatocytes; please replace with comparable images.

Reply: We apologize for not carefully check these images. As suggested, the image of the PBS control group in Fig. 1d has now been replaced with comparable images with central veins and pericentral hepatocytes

Fig. 1. (d) H&E staining of liver sections. Scale bar, 50 μm.

- The entire manuscript should be checked for grammar/typing mistakes; for example, in the limitations of the study section, what are the 'satellite' cells?

Reply: Thanks for your suggestions. In this revision, we carefully checked all these mistakes and made corresponding corrections.

Again, we highly appreciate your insightful comments.

Reviewer #2 (Remarks to the Author):

In this study, the authors suggested a protective effect of ASGR1 against liver injury, and ASGR1 deficiency increases circulating levels of GP73 and subsequently activates ER stress via elevating the interaction between GP73 and BIP, and consequently, leading to liver injury. Moreover, their data also demonstrated that neutralization of GP73 markedly attenuated the liver injury, in mouse model.

Though the work is interesting, there are some concerns need to be addressed:

Reply: We thank the reviewer for the positive comments. We now addressed all of your questions, either by providing clarification, by presenting additional results, or by performing new experiments. Point-by-point responses to your comments are given below.

1. Fibrosis is an important stage prior to cirrhosis, how about ASGR1's expression in fibrotic hepatocytes?

Response: To examine ASGR1's expression in fibrotic hepatocytes, we performed following experiments: (1) We isolated primary hepatocytes from mice with CCl₄-induced fibrosis and compared their ASGR1 expression with that of hepatocytes from normal mice. The results showed reduced mRNA (**Fig. R6a**) and protein (**Fig. R6b, c**) levels of ASGR1 in hepatocytes from mice with liver fibrosis. Additionally, our original results showed significantly downregulated hepatic ASGR1 mRNA and protein levels in CCl₄-treated liver fibrotic mice (**Fig. 1h, i, k**). (2) In liver biopsy specimens from 6 patients with liver fibrosis and 6 normal controls, we found significantly reduced hepatic mRNA and protein levels of ASGR1 from patients with liver fibrosis compared with normal controls (**Fig. S1a-c**). Furthermore, immunofluorescence staining of these liver sections showed decreased protein levels of ASGR1 in patients with liver fibrosis (**Fig. S1d, e**). (3) Analysis of ASGR1 mRNA expression from publicly available GEO database revealed significantly reduced levels in patients with advanced liver fibrosis compared to mild liver fibrosis (**Fig. R6d**). We also analyzed the publicly available single-cell nuclear transcriptome data from patients with liver fibrosis (Data derived from Wang, et al. *Sci Transl Med*, PMID: 36599008) and found markedly decreased ASGR1 expression in hepatocytes from NASH patients with liver fibrosis compared to hepatocytes from healthy individuals (**Fig. R6e**). Collectively, these results demonstrate reduced ASGR1 expression in fibrotic hepatocytes.

Fig. R6. ASGR1 expression is downregulated in fibrotic hepatocytes. (a to c) Primary hepatocytes isolated from mice with CCl₄-induced fibrosis or normal controls. (a and b) Relative mRNA (n=6) and protein (n=3) expression of ASGR1. (c) Quantification of ASGR1 protein levels (n=3). (d) Relative mRNA expression of ASGR1 in patients with mild fibrosis or advanced fibrosis from publicly available GEO database. (e) Relative mRNA expression of ASGR1 in hepatocytes from NASH patients with liver fibrosis or healthy individuals from public scRNA-seq data sets. Data are presented as mean \pm SEM. *P* values were calculated by two-tailed unpaired t-test. **P* < 0.05, ***P* < 0.01.

Fig. 1. (h and i) Relative mRNA and protein expression of hepatic ASGR1 in mice treated with CCl₄. **(k)** Representative immunofluorescence staining of ASGR1 in livers of mice treated with CCl₄. Scale bar, 50 μm. Data are presented as mean ± SEM. *P* values were calculated by two-tailed unpaired t-test. **P* < 0.05, ***P* < 0.01.

Fig. S1. Hepatic ASGR1 expression is downregulated in patients with liver fibrosis. Liver biopsy specimens were collected from 6 patients with liver fibrosis and 6 normal controls. **(a and b)** Relative mRNA and protein expression of ASGR1. **(c)** Quantification of ASGR1 protein levels (n=3). **(d)** Representative immunofluorescence staining of hepatic ASGR1 in patients with liver fibrosis and normal controls (n=3). Scale bar, 25 μ m. **(e)** Quantification of immunofluorescence staining of ASGR1 in patients with liver fibrosis and normal controls. Data are presented as mean \pm SEM. *P* values were calculated by two-tailed unpaired t-test. **P* < 0.05, ***P* < 0.01.

The link of GP73 to liver disease was firstly reported by the finding that is highly expressed in liver cancer. How about GP73's relationship with ASGR1 in the malignant transformed hepatic cells.

Reply: To investigate the relationship between ASGR1 and GP73 in the malignant transformed hepatic cells, we measured cellular expression of ASGR1 and medium levels of GP73 in cultured hepatic cell lines with different malignant potentials (MHCC97H with high malignant potential, Huh7 with moderate malignant potential, HepG2 with low malignant potential) (Jia, et al. *J Exp Clin Cancer Res*, PMID: 28870205, Chuang, et al. *Hepatology*, PMID: 25820676). The results showed that cellular ASGR1 expression was decreased while medium GP73 levels were increased with the increase of malignancy potential (**Fig.R7a, b**). Further analysis showed an inverse correlation between mRNA expression of ASGR1 and medium levels of GP73 in hepatic cell lines (**Fig.R7c**). Given that epithelial-mesenchymal transition (EMT) allows the solid tumors to become more malignant, increasing their invasiveness and metastatic activity, and TGF- β 1 is able to induce the EMT of tumor cells (Xu, et al. *Cell Res*, PMID: 14672557), we treated HCC cell lines with TGF β 1 to promote malignant transformation. Compared with their respective untreated controls, we observed significantly decreased cellular ASGR1 mRNA expression and increased medium GP73 levels in MHCC97H and HepG2 cells treated with TGF- β 1, with higher levels in highly malignant MHCC97H than in low malignant HepG2 cells (**Fig. R7d-h**). Meanwhile, we also found an inverse correlation between ASGR1 mRNA expression and medium levels of GP73 in MHCC97H or HepG2 cells treated with or without TGF- β 1 (**Fig. R7f, i**). Furthermore, we measured hepatic ASGR1 mRNA expression and serum levels of GP73 in patients with hepatocellular carcinoma at different stages. The results showed that the expression of hepatic ASGR1 was decreased while serum GP73 levels were increased with cancer stages (**Fig. S12a-b**), showing a negative correlation between them in patients with hepatocellular carcinoma at different stages (**Fig. S12c**). These data suggest a negative correlation between secreted GP73 protein levels and ASGR1 mRNA levels in the malignant transformed hepatic cells.

Fig. R7. The negative correlation between ASGR1 mRNA expression and secreted GP73 levels in malignant hepatic cells. (a) Relative mRNA expression of ASGR1 in HepG2, Huh7 and MHCC97H cells (n=6). **(b)** GP73 release in HepG2, Huh7 and MHCC97H cells cultured for 24 h (n=6). **(c)** The correlation between ASGR1 mRNA expression and medium GP73 levels analyzed by Pearson's correlation analysis in HepG2, Huh7 and MHCC97H cells. **(d)** Relative mRNA expression of ASGR1 in HepG2 cells treated with or without TGF-β1 (5ng/ml) for 24 h (n=6). **(e)** GP73 release in culture medium of HepG2 cells treated with or without TGF-β1 (5ng/ml) for 24 h (n=6). **(f)** The correlation between ASGR1 mRNA expression and medium GP73 levels analyzed by Pearson's correlation analysis in HepG2 cells treated with or without TGF-β1. **(g)** Relative mRNA expression of ASGR1 in MHCC97H cells treated with or without TGF-β1 (5ng/ml) for 24 h (n=6). **(h)** GP73 release in culture

medium of MHCC97H cells treated with or without TGF- β 1 (5ng/ml) for 24 h (n=6). (i) The correlation between *ASGR1* mRNA expression and medium GP73 levels analyzed by Pearson's correlation analysis in MHCC97H cells treated with or without TGF- β 1. Data are presented as mean \pm SEM. *P* values were calculated by two-tailed unpaired t-test (d, e, g, h), or one-way ANOVA with Tukey's multiple comparisons test (a, b). **P* < 0.05, ***P* < 0.01.

Fig. S12. Hepatic *ASGR1* mRNA expression was negatively correlated with serum levels of GP73 in patients with hepatocellular carcinoma. (a) Relative hepatic mRNA expression of *ASGR1* (n=6). **(b)** Serum levels of GP73 (n=6). **(c)** Correlation between hepatic *ASGR1* mRNA expression and serum GP73 levels in patients with hepatocellular carcinoma. Data are presented as mean \pm SEM. *P* values were calculated by one-way ANOVA with Tukey's multiple comparisons test. **P* < 0.05, ***P* < 0.01.

3. The authors showed that *ASGR1* deficiency increases circulating levels of GP73, and *ASGR1* physically binding to GP73 protein, would *ASGR1* somehow involved in the digestion and release of GP73 into circulation?

Reply: We highly appreciate your fascinating question. Our current data in original Fig. 6b, c showed that *ASGR1* overexpression significantly increased fluorescence intensity in HepG2 cells incubated with fluorescent-labeled recombinant GP73, whereas the employment of a high-affinity natural ligand for *ASGR1* significantly inhibited this process. In addition, treatment with Chloroquine, a lysosome-specific inhibitor, significantly blocked the lysosomal localization of GP73 in *ASGR1*-overexpressing hepatocytes (Fig. 6f, g). These results suggest that *ASGR1*, the cell membrane receptor, is able to ingest serum/medium GP73 via endocytosis and mediates its lysosomal degradation.

Inspired by your insightful comment, we further investigated whether *ASGR1* is also involved in the release of GP73 into the circulation. To this end, we knocked down *CHC* in HepG2 cells to prevent the interference of *ASGR1*-mediated GP73 endocytosis on medium GP73 content. We observed that *ASGR1* overexpression had no effect on the release of GP73 (Fig. S14a, b). It has been shown that GP73 is cleaved by furin before being released into

circulation (Wei, C., et al. *Hepatology*, PMID: 30723919). Consistent with this, increased GP73 secretion was observed when furin was overexpressed in GP73-overexpressing HepG2 cells (Fig. S14c, d). However, the expression of furin in *ASGR1*-overexpressing cells was not significantly different from that in controls (Fig. S14e, f). Altogether, these data indicate that *ASGR1* is involved in the endocytosis and lysosome degradation of GP73, but not the release of GP73 into circulation.

Fig. 6. (b) Immunofluorescence analysis of HepG2 cells with or without overexpression of *ASGR1* incubated with fluorescent-labeled recombinant GP73 (red) in the presence or absence of Asialofetuin A (30 $\mu\text{g}/\text{mL}$). Scale bars, 5 μm . (c) Quantification of the fluorescence intensity of GP73 shown in (b). (f) Immunofluorescence analysis of HepG2 cells with or without *ASGR1* overexpression incubated with fluorescent-labeled recombinant GP73 (red) and anti-LAMP1 antibody (green) in the presence or absence of chloroquine (5 μM). Scale bars, 5 μm . (g) Quantification of GP73-LAMP1 colocalization shown in (f). Data are presented as mean \pm SEM. * $P < 0.05$, ** $P < 0.01$. OE, overexpression; LAMP1, lysosomal-associated membrane protein 1.

Fig. S14. Hepatic ASGR1 is not involved in GP73 release. (a) Medium GP73 levels in control or *ASGR1*-overexpressing HepG2 cells transfected with si-*CHC* (n=3). (b) Relative protein levels of GP73 in cells or medium of control or *ASGR1*-overexpressing HepG2 cells transfected with si-*CHC* (n=3). (c) Medium GP73 levels in control or *GP73*-overexpressing HepG2 cells transfected with or without *Furin* (n=3). (d) Relative protein levels of GP73 in cells or medium of control or *GP73*-overexpressing HepG2 cells in the presence or absence of *Furin* overexpression. (e and f) Relative mRNA and protein expression of *Furin* in control or *ASGR1*-overexpressing HepG2 cells (n=3). Data are presented as mean \pm SEM. *P* values were calculated by two-tailed unpaired t-test (a, e), or one-way ANOVA with Dunnett's multiple comparisons test (c). **P* < 0.05, ***P* < 0.01.

4. Last but important question: the authors reported in the submission that neutralization of GP73 markedly attenuated the liver injury, and GP73 intercellular interacted with BIP to active ER stress. How to explain such outer an inter difference?

Reply: Thanks for your question. Although best known as an ER luminal protein, BIP (GRP78) is also highly expressed on the plasma membrane where it functions as a cell surface signaling receptor (Misra, U. K., et al. *J Biol Chem*, PMID: 7513689, Misra, U. K., et al. *J Biol Chem*, PMID: 12194978). Previous study showed that extracellular GP73 is a signaling molecule in response to ER stress and interacted with the cell surface protein BIP, activating ER stress signaling in hepatocytes (Wei, C., et al. *Hepatology*, PMID: 30723919). By Immunoprecipitation analysis of the cell-surface proteins and immunofluorescence analysis, they showed that extracellular GP73 is able to associate with BIP at the cell membrane of hepatocytes (**Fig 4F, G** from *Hepatology*, PMID: 30723919, also shown below).

Fig. 4F, G from *Hepatology*, **PMID: 30723919**. **(F)** Immunoprecipitation analysis of the cell-surface proteins isolated from HepG2 cells after treatment with Tm (10 rane was incuhrs. **(G)** Representative confocal immunofluorescence images (left) and quantification (right) of GP73 colocalized with GRP78 in the plasma membrane of HepG2 cells after treatment with Tm (10 P78 in escence imagese proteins GP73, green represents GRP78, and blue represents the plasma membrane. Scale bar, 10 re.

To investigate whether ASGR1 regulates the binding of cell-surface protein BIP to GP73, we transfected *ASGR1*-overexpressing or *ASGR1*-knockdown HepG2 cells and performed immunofluorescence analysis for GP73, BIP, and the plasma membrane marker, Na⁺/K⁺-ATPase (Fig. S15c). The results showed that colocalization of GP73 and BIP at the plasma membrane was significantly increased upon *ASGR1* knockdown, while decreased upon *ASGR1* overexpression. Together, these results revealed that GP73 induced hepatic ER stress leading to liver injury by interaction with the cell-surface BIP, and ASGR1 deficiency upregulates the interaction between BIP and GP73 at the plasma membrane, leading to the activation of hepatic ER stress.

Fig. S15. ASGR1 regulates the interaction between BIP and GP73 at the plasma membrane. (c) Representative confocal immunofluorescence images (left) and quantification (right) of GP73 colocalization with BIP at the plasma membrane of *ASGR1*-knockdown or *ASGR1*-overexpressing HepG2 cells. Red represents GP73, green represents BIP, and blue represents the plasma membrane. Scale bars, 5 μm.

Reviewer #3 (Remarks to the Author):

Manuscript Number: NAR-01594-Y-2023

Manuscript Title: Deficiency 1 of ASGR1 promotes liver injury by increasing GP73-mediated hepatic endoplasmic reticulum stress

Genetic factors leading to the initiation and progression of the liver injury remain unknown and it is critical for developing therapeutic intervention. In this report authors evaluated the role of asialoglycoprotein receptor (ASGR1) in liver injury of mice. Authors demonstrate that ASGR1 deficiency exacerbates the liver injury. In addition, they also demonstrate that ASGR1 overexpression improves APAP-induced acute and CCl₄-induced chronic liver injuries in mice. Their studied to understand mechanism reveals that ASGR1 binds GP73 which in turn trigger lysosomal degradation and ER stress play a critical role in liver injury of mice. Authors also found that cirrhotic patients with decreased ASGR1 expression display increased serum GP73 levels and hepatic stress. The finding reported in this manuscript identify ASGR1 as a novel candidate underlying genetic cause to liver injury. This article is well written, and all experiments were executed diligently. This article is suitable for publication in Nature Communications.

Reply: We thank the reviewer for the thoughtful review and acknowledging suitability of the manuscript publication.

REVIEWERS' COMMENTS

Reviewer #1 (Remarks to the Author):

The authors considered my concerns and the revised manuscript is significantly improved.

Reviewer #2 (Remarks to the Author):

Since the revised version has addressed all my concerns, there is no more comments to the submission.